# Reusable Slotwise Mechanisms

**Trang Nguyen**[*]
Mila – Quebec AI Institute
FPT Software AI Center
Montreal, Canada
trang.nguyen@mila.quebec

**Amin Mansouri**
Mila – Quebec AI Institute
Université de Montréal
Montreal, Canada
amin.mansouri@mila.quebec

**Kanika Madan**
Mila – Quebec AI Institute
Université de Montréal
Montreal, Canada
madankan@mila.quebec

**Khuong Nguyen**
FPT Software AI Center
Ho Chi Minh City, Vietnam
khuongnd6@fpt.com

**Kartik Ahuja**
Mila – Quebec AI Institute
Montreal, Canada
kartik.ahuja@mila.quebec

**Dianbo Liu**
Mila – Quebec AI Institute
National University of Singapore
Montreal, Canada
dianbo.liu@mila.quebec

**Yoshua Bengio**
Mila – Quebec AI Institute
Université de Montréal, CIFAR AI Chair
Montreal, Canada
yoshua.bengio@mila.quebec

## Abstract

Agents with the ability to comprehend and reason about the dynamics of objects would be expected to exhibit improved robustness and generalization in novel scenarios. However, achieving this capability necessitates not only an effective scene representation but also an understanding of the mechanisms governing interactions among object subsets. Recent studies have made significant progress in representing scenes using object slots. In this work, we introduce Reusable Slotwise Mechanisms, or RSM, a framework that models object dynamics by leveraging communication among slots along with a modular architecture capable of dynamically selecting reusable mechanisms for predicting the future states of each object slot. Crucially, RSM leverages the *Central Contextual Information (CCI)*, enabling selected mechanisms to access the remaining slots through a bottleneck, effectively allowing for modeling of higher order and complex interactions that might require a sparse subset of objects. Experimental results demonstrate the superior performance of RSM compared to state-of-the-art methods across various future prediction and related downstream tasks, including Visual Question Answering and action planning. Furthermore, we showcase RSM's Out-of-Distribution generalization ability to handle scenes in intricate scenarios.

## 1 Introduction

Accurate prediction of future frames and reasoning over objects is crucial in various computer vision tasks. These capabilities are essential for constructing comprehensive world models in applications like autonomous driving and reinforcement learning for robots. Traditional deep learning-based representation learning methods compress entire scenes into monolithic representations, lacking

---

[*]Work done under an internship at Mila - Quebec AI Institute and FPT Software AI Residency Program. Corresponding author.

37th Conference on Neural Information Processing Systems (NeurIPS 2023).

compositionality and object-centric understanding. As a result, these representations struggle with systematic generalization, interpretability, and capturing interactions between objects. This limitation leads to poor generalization performance as causal variables become entangled in non-trivial ways.

There has been growing interest in slot-based and modular representations that decompose scenes into individual entities, deviating from fixed-size monolithic feature vector representations [19, 38, 16, 17, 15, 18, 32, 21, 28, 37, 39, 48, 30]. These novel approaches offer significantly more flexibility when dealing with environments that comprise multiple objects. By employing an encoder that segments a scene into its independent constituent entities instead of compressing information into a fixed-size representation, these methods allow for greater flexibility and parameter sharing when learning object-centric representations, and their compositional nature enables better generalization. Compositional and object-centric representations can be effectively utilized alongside complex world models that accurately capture the interactions and dynamics of different entities in a scene.

These world models, when presented with proper representations, in principle, can model the transition functions that relate latent causal factors across consecutive time steps of a rollout. While monolithic blocks are still used occasionally with object-centric methods [44], recent attempts have incorporated similar inductive biases related to the object-centricity of images in modeling interactions. Structured world models and representations seem truly promising for systematically generalizing to novel scenes. Structured world models would ideally decompose the description of the evolution of a scene into causal and independent sub-modules, making it easy to recombine and repurpose those mechanisms in novel ways to solve challenges in unseen scenarios. Such separation of dynamics modeling makes structured world models more adaptable to distribution shifts, as only the parameters of a few mechanisms that have changed in a new environment would have to be retrained, and not all of the parameters in the case of a monolithic model [8].

A major class of such structured world models aims at baking in some inductive bias about the nature of object interactions. On one extreme, there have been studies that employ Graph Neural Networks (GNNs) to capture object dependencies through dense connections, while on the other hand, there has been contrasting work aiming at modeling the dynamics through only pairwise interactions. We believe, however, that ideally, an agent should be able to learn, select, and reuse a set of prediction rules based on contextual information and the characteristics of each object.

In this work, we argue that the assumptions made in previous attempts at learning the dynamics among slots may be insufficient in more realistic domains. To address these limitations, we propose Reusable Slotwise Mechanisms (RSM), a novel modular architecture incorporating a set of deep neural networks representing reusable mechanisms on top of slotwise representations [31, 10]. Inspired by the Global Workspace Theory (GWT) in the cognitive neuroscience of working memory [2, 3], we introduce the concept of Central Contextual Information (CCI), which allows each reusable mechanism, *i.e.*, a possible explanation of state evolution, to access information from all other slots through a bottleneck, enabling accurate predictions of the next state for a specific slot. The CCI's bottleneck amounts to a relaxed inductive bias compared to the extreme cases of pairwise or fully dense interactions among slots. Finally, comprehensive experiments demonstrate that RSM outperforms the state-of-the-art in various next-step prediction tasks, including independent and identically distributed (iid) and Out-of-Distribution (OOD) scenarios.

In summary, the presented work makes the following contributions:

1. RSM: A modular dynamics model comprising a set of reusable mechanisms that take as input slot representations through an attention bottleneck and sequential slot updates.

2. RSM strengthens communication among slots in dynamics modeling by relaxing inductive biases in dynamics modeling to be not too dense or sparse but depend on a specific context.

3. RSM outperforms baselines in multiple tasks, including next frames prediction, Visual Question Answering, and action planning in both iid and OOD cases.

## 2 Proposed Method: RSM - Reusable Slotwise Mechanisms

### 2.1 RSM Overview

We introduce RSM, a modular architecture consisting of a set of $M$ Multilayer Perceptrons (MLPs) that act as reusable mechanisms, operating on slotwise representations to predict changes in the slots.

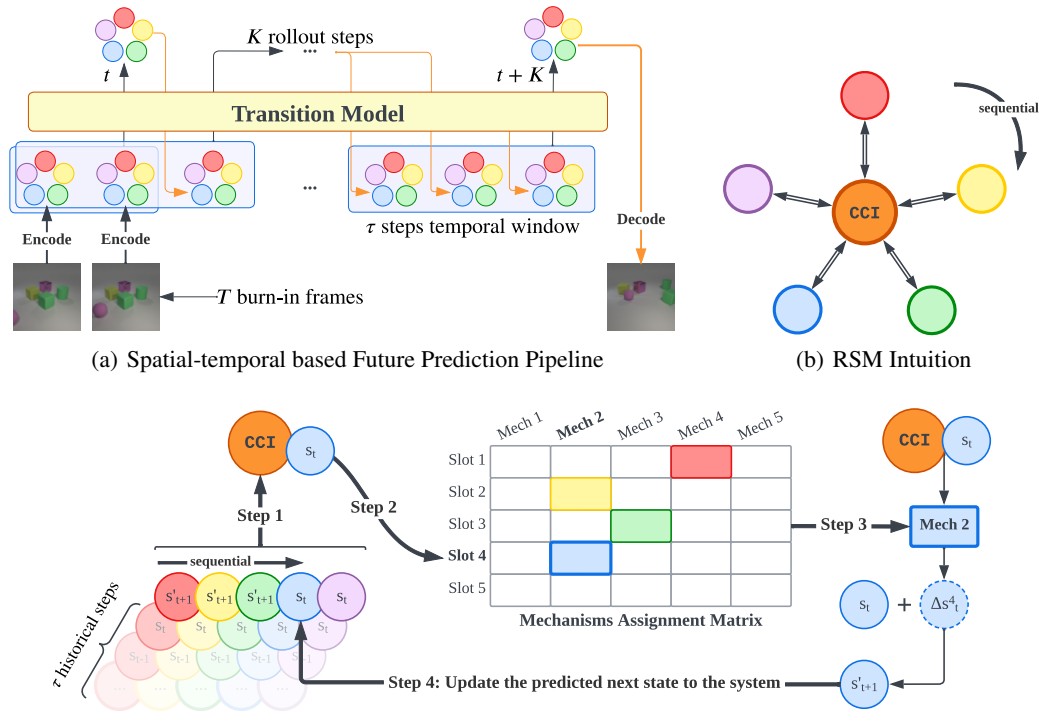

(a) Spatial-temporal based Future Prediction Pipeline

(b) RSM Intuition

(c) Computational Flow in RSM from step $t$ to $t+1$

Figure 1: The future prediction pipeline (Figure 1(a)), RSM Intuition (Figure 1(b)), and Computational Flow (Figure 1(c)). Colored circles represent slots, with the dashed border denoting changes in a slot. The Central Contextual Information (CCI) is derived from all object slots as context, assists in selecting a mechanism for slots, and acts as an input of mechanisms. Slots are sequentially updated in four steps: **(1)** Compute CCI by unrolling slots (updated and non-updated) over the past $\tau$ steps, **(2)** Select a mechanism based on CCI and the slot of interest, **(3)** Predict the next state by the selected mechanism's dynamics, and **(4)** Update the predicted slot and prepare for the next object's turn.

What sets RSM apart from other architectures is incorporating the CCI buffer, which enhances its adaptability when dealing with environments characterized by varying levels of interaction complexity among objects by allowing the propagation of information about all other slots. Unlike previous approaches [18, 16, 17, 23], we enable a sparse subset of slots to transmit contextual information through a bottleneck for updating each slot. This inductive bias becomes helpful in environments where higher-order complex interactions need to be captured by reusable mechanisms as the CCI effectively modulates the complexity of the mechanisms and accommodates those that require one or two slots, as well as those that rely on a larger subset of slots.

The training pipeline of RSM, which is visualized in Figure 1(a), is designed to predict $K$ rollout steps of $N$ object slots, based on $T$ burn-in frames with a temporal window of maximum $\tau$ history steps for each prediction step. Prediction of the rollout begins by processing the given $T$ burn-in frames to obtain the $N$ slot representations for each of the $T$ burn-in steps using an object-centric model. For any rollout step after the burn-in frames, the slots in a window of $\tau \leq T$ previous steps will be fed to the model as additional context to predict the state (slots) at $t+1$. The model takes this input and updates the slots sequentially to output the slots at $t+1$. The sequential updating of slots means that updates from (according to some ordering) slots can influence the prediction of later slots within a prediction step, as illustrated in Figure 1(b). It is worth highlighting that the sequential way of updating slots breaks the symmetries in mechanism selection, and also allows for a more expressive transition function, similar to how autoregressive models enable encoding of rich distributions. The prediction process is repeated until the slots for $K$ rollout steps are predicted.

## 2.2 Computational Flow in RSM

This section describes the computational flow of RSM in more detail along with a 4-step process that will be repeated for all slots within a time-step $t$, in a sequential manner, as illustrated in Figure 1(c) and Algorithm 1 in the Appendix. The following are the main components of the architecture, where $d_s$ and $d_{cci}$ denote the dimension of slots and the CCI, respectively:

1. **MultiheadAttention**$(\cdot)$ : $\mathbb{R}^{((\tau+1)\times N)\times d_s} \to \mathbb{R}^{d_s}$ followed by a **projection** $\phi(\cdot)$ : $\mathbb{R}^{d_s} \to \mathbb{R}^{d_{cci}}$ that computes the CCI, denoted as cci $\in \mathbb{R}^{d_{cci}}$, from all of the $N$ slots in the past $T$ steps concatenated. Keys, queries, and values all come from slots, so the CCI is not affected by the order of slot representations.

2. The set of $M$ **reusable mechanisms** $\{g_1, \ldots, g_M\}$ where $g_i(\cdot) : \mathbb{R}^{d_{cci}+d_s} \to \mathbb{R}^{d_s}$ are represented by independently parametrized MLPs implicitly trained to specialize in explaining different transitions. Each such $g_i(\cdot)$ takes as input one slot concatenated with the CCI.

3. $\psi(\cdot) : \mathbb{R}^{d_{cci}+d_s} \to \mathbb{R}^M$ that takes as input the CCI and the slot of interest $s_t^i$. It computes a categorical distribution over the possible choices of mechanisms for $s_t^i$, and outputs a sample of that distribution to be used for updating $s_t^i$.

Considering the $N$ slots per each of the $\tau$ steps in the temporal window before $t$, $s_{\tau^*:t}^{1:N} = \{s_{t-\tau+1}^1, s_{t-\tau+1}^2, \ldots, s_{t-\tau+1}^N, \ldots, s_t^1, s_t^2, \ldots, s_t^N\}$ with $\tau^* = t - \tau + 1$, RSM predicts the next state of slots, denoted as $s_{t+1}'^{1:N}$, using the following 4-step process, which is sequentially applied to each of the slots. Suppose an ordering has been fixed over the slots for a rollout, and according to that ordering, for some $0 < n \leq N$, we have that $n - 1$ slots have been updated to their predicted values at $t + 1$ and are denoted by $s_{t+1}'^1, \ldots, s_{t+1}'^{n-1}$. Below, we explain the process of computing the next state for $s_t^n$ (e.g. the blue slot in Figure 1(c)).

***Step 1.*** **Compute the CCI**: We first append $s_t'^{1:N}$ to the current temporal window to achieve $s_{\tau^*:t+1}^{1:N}$. The duplicated $s_t'^{1:N}$ serves as a placeholder, which will be overwritten with the predicted values in subsequent steps. Subsequently, as presented in Equation 1, a **MultiheadAttention**$(\cdot)$ takes $s_{\tau^*:t+1}^{1:N}$ as input before passing through $\phi(\cdot)$ to produce the central contextual information cci.

$$\text{cci} = \phi(\textbf{MultiheadAttention}(s_{\tau^*:t+1}^{1:N})) \tag{1}$$

***Step 2.*** **Select one mechanism for $s_t^n$**: $\psi(\cdot)$ takes two arguments as inputs, cci from Step 1 and $s_t^n$, to output the logits of a categorical distribution $\pi_{1:M}$ over $M$ possible choices of mechanisms. We employ a hard Gumbel-softmax layer with Straight-through Trick [33, 22] on top of $\psi$'s outputs, as described in Equation 2, to select the mechanism.

$$\pi_{1:M} = \text{Gumbel-softmax}(\psi(\text{cci}, s_t^n)) \tag{2}$$

***Step 3.*** **Predict the changes of $s_t^n$**: Let $\Delta s_t^n$ denote the change of $s_t^n$ from $t$ to $t + 1$. $\Delta s_t^n$ is presented in Equation 3, where $k = \text{argmax}(\pi_{1:M})$.

$$\Delta s_t^n = g_k(\text{cci}, s_t^n) \tag{3}$$

***Step 4.*** **Update and sync $s_{t+1}^n$**: $s_{t+1}'^n$ is then computed by adding the predicted transformation from the previous step and replaces the value of $s_{t+1}^n$ in the slots buffer, as described in Equation 4.

$$s_{t+1}'^n = s_t^n + \Delta s_t^n \tag{4}$$

The process above is repeated for all slots at time $t$, to obtain the next state (slots) prediction $s_{t+1}'^{1:N}$.

## 3 Experiments Setup

This study evaluates RSM's dynamics modeling and generalization capabilities through video prediction, VQA, and action planning tasks. We aim to provide empirical evidence supporting the underlying hypotheses that guided the architectural design of RSM.

- $\mathcal{H}_1$: Slots communication through the CCI and reusable mechanisms reduces information loss during prediction, resulting in accurate prediction of future rollout frames (**Section 4.1**).

- $\mathcal{H}_2$: RSM effectively handles novel scenarios in the downstream tasks (**Section 4.2**), especially enhancing OOD generalization (**Section 4.3**).

- $\mathcal{H}_3$: The synergy between the CCI and the disentanglement of objects dynamics into reusable mechanisms is essential to RSM (analyses in **Section 4.4** and ablations in **Section 4.5**).

In the following subsections, we describe the experiments focusing on the transition of slots over rollout steps with pre-trained object-centric models. Additionally, Appendix E provides experiments and analyses with an end-to-end training pipeline.

## 3.1 Environments

**OBJ3D** [29] contains dynamic scenes of a sphere colliding with static objects. Following Lin et al. [29], Wu et al. [44], we use 3 to 5 static objects and one launched sphere for interaction.

**CLEVRER** [45] shares similarities with OBJ3D, but additionally has multiple moving objects in various directions throughout the scene. For the VQA downstream task, CLEVRER offers four question types: descriptive, explanatory, predictive, and counterfactual, among which, in the spirit of improving video prediction, we focus on boosting the performance on answering predictive questions which require an understanding of future object interactions.

**PHYRE** [4] is a 2D physics puzzle platform where the goal is strategically placing red objects such that the green object touches the blue or purple object. Bakhtin et al. [4] design *templates* that describe such tasks with varying initial states. Subsequently, they define (1) *within-template* protocol where the test set contains the same *templates* as in training, and (2) *cross-template* protocol that tests on different *templates* than those in training. We report results both on *within-template* as iid and on *cross-template* to obtain the OOD evaluation.

**Physion** [7] is a VQA dataset that assesses a model's capability in predicting objects' movement and interaction in realistic simulated 3D environments in eight physical phenomena.

For further details and data visualization, we refer the readers to Appendix B.

## 3.2 Baselines

We compare RSM against three main baselines: **SlotFormer** [44], Switch Transformer [14] denoted as **SwitchFormer**, and **NPS** [17]. We show the efficacy of relaxing the inductive bias on communication density among slots in RSM by contrasting with dense communication methods (SlotFormer and SwitchFormer) and a pair-wise communication method (NPS). Likewise, comparing RSM to SlotFormer highlights the role of disentangling objects' dynamics into mechanisms, while comparing to SwitchFormer and NPS emphasizes the vital role of communication among slots via the CCI. Additionally, we compare to **SAVi-Dyn** [44], which is the SOTA on CLEVRER. In other experiments, we compare to **SlotFormer** (current SOTA).

In the tables, we present our reproduced SlotFormer (marked by "†") and our re-implemented SwitchFormer and NPS (marked by "*"), alongside SAVi-Dyn reported by Wu et al. [44]. [2]

## 3.3 Implementation Details

Following Wu et al. [44], we focus on the transition of slots and take advantage of the pre-trained object-centric *encoder-decoder* pair that convert input frames into slots and vice versa. We use the pre-trained weights of SAVi and STEVE provided by Wu et al. [44], including **SAVi** [26] for OBJ3D, CLEVRER, and PHYRE; and **STEVE** [40] for Physion.

We present the best validation set configuration of RSM for each dataset, along with fine-tuning results and model size scaling in Appendix D. In summary, (1) OBJ3D and CLEVRER include 7 mechanisms, while PHYRE and Physion use 5, and (2) the number of parameters in RSM is slightly lower than that of SlotFormer in corresponding experiments. Additionally, we re-implemented SwitchFormer and NPS with a similar number of parameters as in RSM and SlotFormer.

---

[2]We adapt the code for Switch Transformer and NPS to ensure consistency of experimental setups and evaluations with SlotFormer and RSM (See Appendix C).

Table 1: **Future frame prediction quality on OBJ3D and CLEVRER.** Bold scores indicate the best performance, with the RSM consistently outperforming baselines by a remarkable margin.

| Method | OBJ3D | | CLEVRER | | | | |
|---|---|---|---|---|---|---|---|
| | SSIM↑ | LPIPS$_{\times 100}$↓ | SSIM↑ | LPIPS$_{\times 100}$↓ | ARI↑ | FG-ARI↑ | FG-mIoU↑ |
| SAVi-Dyn | 0.91 | 12.00 | 0.89 | 19.00 | 8.64 | 64.32 | 18.25 |
| NPS* | $0.90^{\pm 0.2}$ | $8.24^{\pm 0.2}$ | $0.89^{\pm 0.2}$ | $12.51^{\pm 0.0}$ | $62.84^{\pm 0.2}$ | $64.62^{\pm 0.3}$ | $30.39^{\pm 0.2}$ |
| SwitchFormer* | $0.91^{\pm 0.2}$ | $8.09^{\pm 0.3}$ | $0.88^{\pm 0.3}$ | $14.28^{\pm 0.1}$ | $60.61^{\pm 0.4}$ | $59.32^{\pm 0.3}$ | $28.94^{\pm 0.2}$ |
| SlotFormer† | $0.90^{\pm 0.2}$ | $8.32^{\pm 0.2}$ | $0.88^{\pm 0.2}$ | $13.09^{\pm 0.1}$ | $63.38^{\pm 0.3}$ | $62.91^{\pm 0.2}$ | $29.68^{\pm 0.3}$ |
| **RSM (Ours)** | $\mathbf{0.92}^{\pm 0.1}$ | $\mathbf{7.88}^{\pm 0.1}$ | $\mathbf{0.91}^{\pm 0.1}$ | $\mathbf{11.96}^{\pm 0.1}$ | $\mathbf{67.72}^{\pm 0.2}$ | $\mathbf{66.15}^{\pm 0.2}$ | $\mathbf{32.73}^{\pm 0.2}$ |

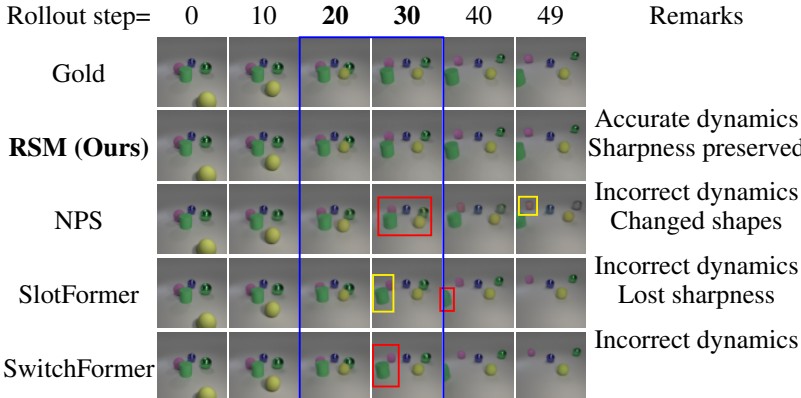

Figure 2: **Comparison of rollout frames in OBJ3D**. RSM renders frames with precise dynamics and holds visual quality, even complex actions in steps 20 to 30. In contrast, the baselines produce ones with artifacts (yellow boxes) and inaccurate dynamics (red boxes). Best viewed in video format.

## 4 Experimental Results

We report mean and standard deviation across 5 different runs. Video visualizations of our experiments are provided in our repository [3]. See also Appendix A on the reproducibility of our results.

### 4.1 Video Prediction Quality

To demonstrate $\mathcal{H}_1$, we provide the video prediction quality on OBJ3D and CLEVRER in Table 1 and Figure 2. In general, RSM demonstrates its effectiveness in handling object dynamics in the long-term future prediction over baselines.

**Evaluation Metrics** We evaluate the quality and similarity of the predicted rollout frames using **SSIM** [42] and **LPIPS** [46] metrics. Since the range of **LPIPS** metric is small, we report the actual values times 100, denoted as **LPIPS**$_{\times 100}$, to facilitate comparisons among the methods. Additionally, we also assess the performance using **ARI**, **FG-ARI**, and **FG-mIoU** metrics, which measure clustering similarity and foreground segmentation accuracy of object bounding boxes and segmentation masks. We evaluate the model's performance averaged over unrolling 44 steps on OBJ3D and 42 steps on CLEVRER with six burn-in frames in both datasets.

Table 1 exhibits that RSM outperforms other approaches and achieves the highest scores across evaluation metrics for both datasets. Notably, compared to SlotFormer, RSM improves LPIPS$_{\times 100}$ by 0.46 points in OBJ3D, 1.12 points in CLEVRER, and increases FG-mIoU by 3.05 points in CLEVRER. Following RSM, NPS consistently ranks second in performance among the baselines.

These results are supported by Figure 2, which illustrates the rollout frames in OBJ3D. RSM's outputs' accurate rollout predictions with high visual fidelity demonstrate the efficacy of slot communication by having less error accumulated along time steps than any of the baselines. It is worth emphasizing that RSM excels in handling a significant series of complex movements in steps 20-30, particularly

---

[3]github.com/trangnnp/RSM

Table 2: **VQA performance on CLEVRER and Physion.** Despite not surpassing human performance in Physion, RSM outperforms baselines in both datasets. All scores are in percentage.

| Method | CLEVRER | | Physion | | |
|---|---|---|---|---|---|
| | per opt. ↑ | per ques. ↑ | Obs. ↑ | Dyn. ↑ | Gap ↑ |
| Human | - | - | **74.7** | - | - |
| NPS* | $95.3^{\pm 0.3}$ | $93.8^{\pm 0.2}$ | | $65.6^{\pm 0.3}$ | +0.6 |
| SwitchFormer* | $92.8^{\pm 0.3}$ | $90.4^{\pm 0.2}$ | 65.0 | $66.2^{\pm 0.1}$ | +1.2 |
| SlotFormer† | $96.1^{\pm 0.2}$ | $93.3^{\pm 0.1}$ | | $66.9^{\pm 0.2}$ | +1.9 |
| **RSM (Ours)** | $\mathbf{96.8}^{\pm 0.1}$ | $\mathbf{94.3}^{\pm 0.0}$ | | $\mathbf{68.1}^{\pm 0.0}$ | **+3.1** |

Table 3: **Action planning task in PHYRE.** RSM outperforms all baselines in both iid and OOD.

| PHYRE-B | NPS* | SwitchFormer* | SlotFormer† | **RSM** |
|---|---|---|---|---|
| **iid** (*within-template*) | $80.52^{\pm 1.0}$ | $78.27^{\pm 1.9}$ | $76.4^{\pm 1.1}$ | $\mathbf{82.89}^{\pm 0.6}$ |
| **OOD** (*cross-template*) | $42.63^{\pm 1.3}$ | $48.36^{\pm 1.4}$ | $42.46^{\pm 1.7}$ | $\mathbf{57.37}^{\pm 1.4}$ |

during a sequence of rapid collision of object pairs. In contrast, we find that the baselines struggle with complex object movements during this period, leading to inaccuracies in predicting the dynamics towards the end. RSM effectively maintains visual quality by predicting the *changes* of slots instead of predicting the next state of a slot. This approach allows RSM to handle action-free scenarios well and significantly reduce error accumulation by facilitating null transitions that preserve slot integrity.

Furthermore, RSM demonstrates flexible slot communication with relaxed inductive biases on interaction density, enabling it to adapt to environments comprising mechanisms with varying levels of complexity. In contrast, we find that NPS with sparse interactions faces difficulties with close-by objects and rapid collisions (seen in OBJ3D), while SwitchFormer with dense communication struggles with distant objects (as in CLEVRER).

## 4.2 Downstream Tasks: Visual Question Answering and Action Planning

### 4.2.1 Visual Question Answering task

To demonstrate $\mathcal{H}_2$, we validate the performance of future frames generated by the models on the VQA task in CLEVRER and Physion, and the action planning task in PHYRE in the next section. The general pipeline is to solve the VQA task with the predicted rollout frames from the given input frames. Specifically, we employ **Aloe** [11] as the base reasoning model on top of the unrolled frames in CLEVRER. Likewise, in Physion, we adhere to the official protocol by training a linear model on the predicted future slots to detect objects' contact. In Physion, we also include the results obtained from human participants [7] for reference. Likewise, we collect the results from observed frames (*Obs.*), which are obtained by training the VQA model on top of pre-trained burn-in slots and compare them to the performance of rollout slots (*Dyn.*).

In Table 2, RSM consistently outperforms all three baselines in VQA for CLEVRER and Physion. On CLEVRER, RSM achieves the highest scores of $96.8\%$ (per option) and $94.3\%$ (per question), surpassing SlotFormer and NPS. In Physion, RSM shows notable improvement, with a $3.1\%$ increase from $65.0\%$ in *Obs.* to $68.1\%$ in *Dyn.*, outperforming all other baselines, indicating the benefit of enhancing the dynamics modeling to improve the downstream tasks. However, RSM is still far from human performance in Physion, showing room for further research into this class of algorithms.

### 4.2.2 Action Planning task

We adopt the approach from prior works [36, 44] and construct a task success classifier as an action scoring function. This function is designed as a simple linear classifier, which considers the predicted object slots, to rank a pre-defined set of 10,000 actions from [4], executed accordingly. We utilize AUCCESS, which quantifies the success rate over the number of attempts curve's Area Under Curve

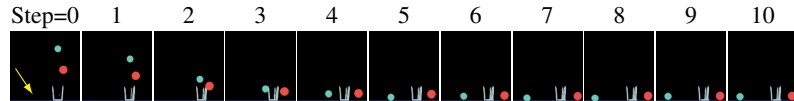

Figure 3: **Action planning task in PHYRE**. RSM strategically positions a red ball at step 0 to prevent the green ball from falling onto the glass by causing a collision that alters the original trajectory of the green ball and causes it to make contact with the blue floor (indicated by the arrow).

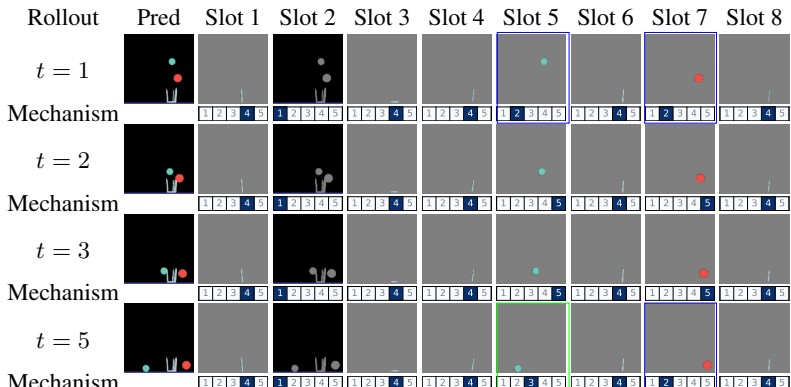

Figure 4: **The underlying mechanism assignment in PHYRE**. Mechanisms are assigned to each slot at $t$ to obtain the updates at $t+1$. RSM disentangles objects' dynamics into reusable mechanisms, which can be expressed as Collision (2), Moving left or right (3), Idle (4), and Falling (5).

(AUC) for the first 100 actions. We report the average score over ten folds, with the best performance among five different runs on each fold.

The first line in Table 3 indicates the action planning results of the proposed RSM and baselines in the iid setting (*within-template* protocol). RSM achieves the highest performance compared to baselines, indicating the critical role of efficient slots communication in complex tasks like action planning. In addition, Figure 3 shows a successful case of RSM solving the planning task by strategically placing the red object at step 0, causing a collision between the green and blue objects at the end.

### 4.3 Out-of-Distribution (OOD) Generalization

To provide more evidence for $\mathcal{H}_2$, we resume analyzing the performance of the action planning task in PHYRE in two OOD scenarios:

**OOD in Templates** Performance on *cross-template* protocol is indicated in the last line in Table 3. Overall, RSM demonstrates strong generalization and has the smallest gap between iid and OOD performance compared to the baselines. The *cross-template* is a natural method to evaluate the OOD generalization in PHYRE since scenes in the train and test sets are in distinct *templates* and contain dissimilar object sizes and objects in the background [4]. We refer the reader to Appendix B for further details and visualizations.

**OOD in Dynamics** We introduce modifications to PHYRE dataset generation that introduce distribution shifts to the dynamics of the blue objects. Particularly, the blue objects are static floors during training. However, in the test set, the object can be assigned as a blue ball, inheriting all dynamics as a red or green ball. As a result, the blue ball appearing only in the test set is considered a strong OOD case, as the model does not have a chance to construct the blue ball and its dynamics during training. Presented in Figure 5, although revealing some distortion in shape, RSM demonstrates the ability to generalize the movements of the ball-shaped object in intensive OOD cases.

### 4.4 The Ability to Disentangle Object Dynamics into Mechanisms

To demonstrate $\mathcal{H}_3$, we conduct qualitative analysis on the underlying mechanisms assignment within the four steps of Figure 3 and visualize results in Figure 4. While there is no explicit assignment

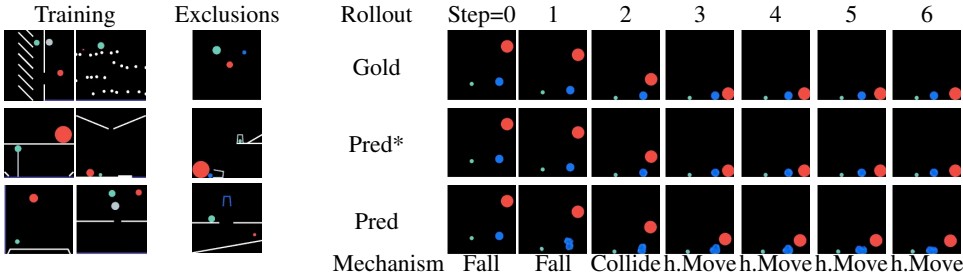

Figure 5: **Ablation on OOD objects' dynamics**. The blue objects are static during training (blue floor) and can move during testing time (blue balls). RSM responds correctly to the blue ball's dynamics. **Pred\*** are predictions by a model trained with the standard training set, **Pred** are predictions by a model trained with the modified training data, and **Mechanism** are mechanisms assigned for the blue object's slot when obtaining **Pred**. **h.Move** denotes the horizontal movement.

| | Gold | RSM | RSM$_{!2}$ | RSM$_{!3}$ | RSM$_{\bar{k}}$ | RSM$_p$ |
|---|---|---|---|---|---|---|
| **LPIPS$_{\times 100}\downarrow$** | - | **7.88** | 15.32 | 9.36 | 13.72 | 8.43 |
| **Step 10** | | | | | | |
| **Step 49** | | | | | | |

Figure 6: **Ablation studies in OBJ3D.** The original design of RSM always demonstrates the superior performance and the accurately predicted future frames compared to the modified versions, including breaking the usages of the CCI in step 2 (**RSM$_{!2}$**) and step 3 (**RSM$_{!3}$**), randomly selecting mechanisms (**RSM$_{\bar{k}}$**), and parallel update of slots (**RSM$_p$**). Best viewed in video format (See our repository).

of roles to mechanisms during training, we can infer their functionality at convergence based on slot visualizations and patterns of activation as follows: Mechanism **2** handles collisions, which can be observed in the collision between the green and red balls in step 1 and that of the red ball with the right-side wall in step 5 (blue boxes). Mechanism **3** controls the horizontal movements, observed in the green ball from steps 5 to 10 (green boxes). Mechanism **4** acts as an idleness mechanism. Mechanism **5** handles downward free-fall motion, observed from steps 2 to 3 of the two balls. Mechanism **1** does not seem to be doing anything meaningful, and this could be due to the model using more capacity than it requires to model the dynamics in this environment.

The inferred functions provide the following insights into understanding the efficacy of RSM. Firstly, RSM successfully disentangles the dynamics into reusable mechanisms, as described in Section 2.1. Secondly, RSM properly assigns such mechanisms to slots throughout the rollout steps, which not only helps preserve the accuracy of prediction but also emphasizes the effectiveness of communication among slots in deciding mechanisms for one another. The automatic emergence of a null mechanism (mechanism 4) is also worth highlighting, which significantly helps reduce the error accumulation in action-free settings, such as idle objects in the background.

## 4.5 Ablation Studies

To provide more evidence for $\mathcal{H}_3$, we conduct ablations to understand the individual effects of (1) the CCI, (2) mechanisms and their disentanglement, and (3) the sequential update of slots in RSM, and visualize the results in Figure 6. Overall, the ablations confirm the superiority of the original RSM design compared to all variations, highlighting a significant disparity in the absence of CCI.

**RSM$_{!2}$** masks out the CCI in step 2 at inference. We observe that the lack of CCI leads to a degenerate selection of mechanisms for slots, with 4 out of 5 object slots being controlled by mechanism 3. **RSM$_{!3}$** masks out the CCI in step 3 during inference. We have found that CCI not only captures comprehensive spatial-temporal information but also provides guidance regarding specific movement

details, including directions. In particular, even when the correct mechanism is assigned (*e.g.*, moving backwards), the slots become confused about the exact direction of movement. Additionally, slots encountered a color-related issue in the subsequent steps.

$\mathbf{RSM_{\bar{k}}}$ randomly assigns mechanisms to slots by a randomized mechanism index, $k$, to replace the distribution $\pi$ in Equation 2. We observe that the launched ball moves in the wrong direction and exits the scene early, while other objects shake in their positions, underscoring the importance of correctly assigning mechanisms to slots.

$\mathbf{RSM_p}$ is the parallel version of RSM that modifies the model $\psi(\cdot)$ in Equation 2 to assign mechanisms to $N$ slots simultaneously, in both training and inference time. We find that $\mathbf{RSM_p}$ stands out as a promising model, as it demonstrates a notable **LPIPS** performance; however, it is essential to note the partially inaccurate dynamics caused by the weaker communication among slots.

## 5 Related Work

**Modular dynamics models.** In the domain of modular neural networks, RIMs [18] pioneered the exploration of modularity for learning dynamics and long-term dependencies in non-stationary settings. However, RIMs suffer from conflating object and mechanism concepts, limiting their effectiveness. SCOFF [16] introduced object files (OF) and schemata to address this limitation, but it struggles with generalizing out-of-distribution (OOD) scenarios. Key distinctions between RSM and SCOFF are as follows: SCOFF schemas can handle only one OF at a time, while RSM allows multiple slots to be input to reusable mechanisms using an attention bottleneck. SCOFF and RIMs use sparse communication among OFs for rare events involving multiple objects, whereas RSM leverages the CCI to activate suitable reusable mechanisms. NPS [17] incorporates sparse interactions directly into its modular architecture, eliminating the need for sparse communication among slots or OFs. Their "production rules" handle rare events involving multiple objects by taking one primary slot and a contextual slot as input. A recent benchmark [23] evaluates causal discovery models and introduces a modular architecture with dense object interactions, similar to GNN-based methods, but assigns separate mechanisms to each object. Among the class of algorithms with less modularity in their dynamics model, R-NEM [41] was a pioneering unsupervised method for modeling dynamics using a learned object-centric latent space through iterative inference (see also [13]) which along similar approaches ([6], [5]) used Graph Neural Networks (GNNs) to model pairwise interactions and differ from our work in two key aspects. Firstly, we do not rely on GNNs to model interactions, as dense interactions are not always realistic in many environments. Secondly, we focus on learning a set of simple and reusable mechanisms that can be applied flexibly to different scenarios, rather than compressing information through shared node and edge updates in a GNN.

**Unsupervised learning of object-centric representations.** To decompose a scene into meaningful sub-parts, there have been lots of recent works on unsupervised learning of object-centric representations from static images [48, 20, 31, 47], videos [35, 27, 25, 12, 44], and theoretical results on the identifiability of such representations [34, 1, 9]. Although lots of these methods work very well in practice, we decided to proceed with slot attention [31] to be consistent with the baselines.

## 6 Conclusion

In this study, we developed RSM, a novel framework that leverages an efficient communication protocol among slots to model object dynamics. RSM comprises a set of reusable mechanisms that take as input slot representations passed through a bottleneck, the Central Contextual Information (CCI), which are then processed sequentially to obtain slot updates. Through comprehensive empirical evaluations and analysis, we show RSM's advantages over the baselines in various tasks, including video prediction, visual question answering, and action planning tasks, especially in OOD settings. Our results suggest the importance of CCI, which integrates and coordinates knowledge from different slots for both mechanism assignment and predicting slot updates. We believe there is a promise for future research endeavors in exploring more sophisticated stochastic attention mechanisms for information integration, aligning with the principles of higher-level cognition, coping with a large number of object slots, and enabling uncertainty quantification in the predictions. In Appendix F, we discuss the limitations of this work and future directions.

## Acknowledgement

We gratefully acknowledge the support received for this research, including from Samsung, CIFAR and NSERC. This research was partly enabled by computational resources provided by Mila - Quebec AI Institute and The Digital Research Alliance of Canada (formerly known as Compute Canada). Each member involved in this research is funded by their primary institution. Amin Mansouri acknowledges support from Microsoft. Additionally, we sincerely thank the anonymous reviewers for their valuable comments and contributions to improve this work.

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

# A    Reproducibility

Each experiment is trained on 4 V100-GPUs with 12 CPUs, using a distributed data-parallel training strategy. The number of parameters and the average training time over 5 runs are summarized in Table 4. In addition, Table 5 provides the necessary configurations to reproduce our work, including the setting related to datasets and the training process that follows the prior work [44], and RSM's design that achieves the best tuning results.

Regarding the rollout frames $K$ and the video length $V$ in Table 5, we predict and consider $K$ future frames from the last burn-in steps for training, whereas, we produce the total of $V$ frames in the inference time, including $T$ burn-in and $V - T$ rollout steps. In other words, $V$ equals $T$ plus the actual rollout frames at inference time. Regarding the training process, we employ the Adam optimizer with an initial learning rate of $2 \times 10^{-4}$ and employ a cosine decay schedule to 0. We further discuss the RSM design in Appendix C.

Table 4: Summary of the number of parameters and training duration. "M" stands for *millions*. "h" stands for GPU hours.

|  | RSM | NPS | SwitchFormer | SlotFormer |
|---|---|---|---|---|
| **OBJ3D** | | | | |
| Num. Params | 0.76M | 0.99M | 0.82M | 0.82M |
| Training Duration | 21h | 25h | 20h | 21h |
| **CLEVRER** | | | | |
| Num. Params | 3.1M | 4.06M | 3.22M | 3.22M |
| Training Duration | 82h | 94h | 72h | 86h |
| **PHYRE** | | | | |
| Num. Params | 5.13M | 5.98M | 6.38M | 6.38M |
| Training Duration | 29h | 33h | 28h | 30h |
| **Physion** | | | | |
| Num. Params | 5.61M | 6.7M | 6.41M | 6.41M |
| Training Duration | 32h | 41h | 30h | 34h |

Table 5: Summary of experiments' configuration, including the configuration of datasets, training process, and RSM.

|  | OBJ3D | CLEVRER | Physion | PHYRE |
|---|---|---|---|---|
| Frame Size | $64 \times 64$ | $64 \times 64$ | $128 \times 128$ | $128 \times 128$ |
| Num. Slots $N$ | 6 | 7 | 6 | 8 |
| Slot Size $d_s$ | 128 | 128 | 192 | 128 |
| Burn-in Frames $T$ | 6 | 15 | 15 | 1 |
| Temporal Window $\tau$ | 6 | 15 | 15 | 6 |
| Rollout Steps $K$ | 10 | 10 | 10 | 10 |
| Video Length $V$ | 6+44 | 15+42 | 15+35 | 1+14 |
| Batch Size | 128 | 128 | 128 | 64 |
| Num. Epochs | 200 | 80 | 25 | 50 |
| Object-centric Model | SAVi | SAVi | STEVE | SAVi |
| Loss Weight $\lambda$ | 1.0 | 1.0 | 0.0 | 0.0 |
| Num. Mechanisms $M$ | 7 | 7 | 5 | 5 |
| Num. Layers of $\psi(\cdot)$ | 1 | 2 | 2 | 2 |
| Num. Layers of Mechanism | 1 | 3 | 3 | 3 |

# B    Dataset Collection

We follow the collection and pre-processing of the dataset, including video length and the dataset splits, as done by Wu et al. [44]. In addition, we provide datasets visualizations in Figure 7 and the visualization of *templates* in PHYRE in Figure 8.

**Algorithm 1 Reusable Slotwise Mechanisms**

$N$: number of slots
$M$: number of mechanisms
$\tau$: number of history length
$T$: number of burn-in frames
$K$: number of rollout steps

$d_s$: slot dimension
$d_{cci}$: central contextual information dimension

***Input***: $s_{\tau^*:t}^{1:N} = \{s_{t-\tau+1}^1, s_{t-\tau+1}^2, \ldots, s_{t-\tau+1}^N, \ldots, s_t^1, s_t^2, \ldots, s_t^N\} \in \mathbb{R}^{(\tau \times N) \times d_s}$ with $\tau^* = t - \tau + 1$: unrolled $N$ slots of the previous $\tau$ steps to time $t$
***Output***: $s_{T+1:T+K}^{1:N}$: predicted $N$ slots in the next $K$ steps from time $T+1$ to $T+K$

**Variables in RSM**

- cci $\in \mathbb{R}^{d_{cci}}$: the Central Contextual Information (CCI).
- $s_t^n \in \mathbb{R}^{d_s}$: the slot of interest, which is slot $n^{th}$ at time $t$.
- $p \in \mathbb{R}^M$: the Gumbel distribution over $M$ choices of selecting mechanisms.
- $\Delta s_t^n \in \mathbb{R}^{d_s}$: changes of the $n$-th slot from time $t$ to $t+1$.

**Components in RSM**:

- $\mathbf{W_q}, \mathbf{W_k}, \mathbf{W_v} : \mathbb{R}^{d_s} \to \mathbb{R}^{d_s}$ denote query, key, and value projection layers transforming the unrolled $s_{\tau^*:t}^{1:N}$ in the attention mechanism.
- $\mathbf{MultiheadAttention}(\cdot) : \mathbb{R}^{((\tau+1) \times N) \times d_s} \to \mathbb{R}^{d_s}$: apply self-attention on the $s_{\tau^*:t}^{1:N}$
- $\phi(\cdot) : \mathbb{R}^{d_s} \to \mathbb{R}^{d_{cci}}$ computes the central contextual information by passing the outputs of the $\mathbf{MultiheadAttention}(\cdot)$ through a nonlinear transformation (MLP).
- $\psi(\cdot) : \mathbb{R}^{d_{cci}+d_s} \to \mathbb{R}^M$ computes the unnormalized probability of selecting a mechanism from $M$ possible choices by taking the CCI and slot of interest as input and feeding that to an MLP.
- Set of $M$ mechanisms $g_j(\cdot) : \mathbb{R}^{d_{cci}+d_s} \to \mathbb{R}^{d_s}, j \in \{1 \ldots M\}$: predict the changes of each slot based on the CCI and current state of the slot. These are also realized with MLPs.

**for each** $t$ in $[T \ldots t+K)$ **do**
    ***Step 0***: *Prepare slots buffer*
    $s_{\tau^*:t}^{1:N} = \text{concat}(s_{\tau^*:t}^{1:N}, s_t^{1:N}) \in \mathbb{R}^{((\tau+1) \times N) \times d_s}$

    **for each** $s_t^n$ in $s_t^{1:N}$ with $n \in 1 \ldots N$ **do**
        ***Step 1***: *Compute the central context*
        $\text{cci} = \phi(\mathbf{MultiheadAttention}(\mathbf{W_q}(s_{\tau^*:t+1}^{1:N}), \mathbf{W_k}(s_{\tau^*:t+1}^{1:N}), \mathbf{W_v}(s_{\tau^*:t+1}^{1:N})))$

        ***Step 2***: *Select a mechanism for slot $s_t^n$*
        $p = \text{Gumbel-max}(\psi(\text{concat}(\text{cci}, s_t^n)))$

        ***Step 3***: *Apply the selected mechanism to slot $s_t^n$. Note that $p$ is one-hot-like distribution.*
        $\Delta s_t^{n,j} = g_j(\text{concat}(\text{cci}, s_t^n)) * p^j \quad \forall j \in \{1, \ldots, M\}$
        $\Delta s_t^n = \sum_{j=1}^M \Delta s_t^{n,j}$

        ***Step 4***: *Update the slots buffer with the new value of $s_{t+1}^n$*
        $s_{t+1}^n = s_t^n + \Delta s_t^n$
    **end for**
**end for**
**return** $s_{T+1:T+K}^{1:N}$

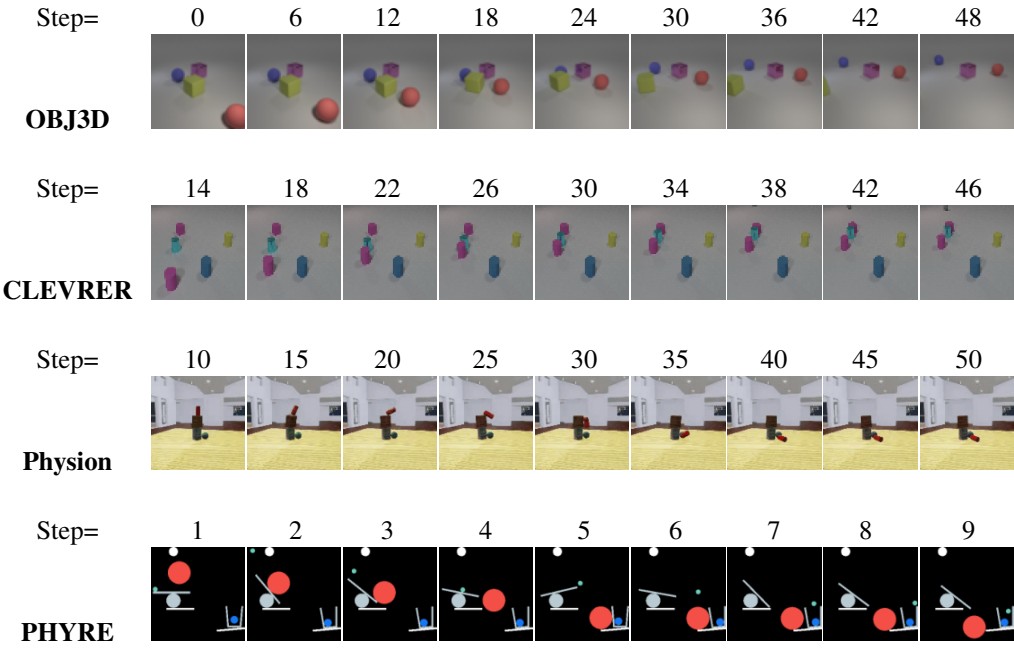

Figure 7: **Dataset visualization over steps.**

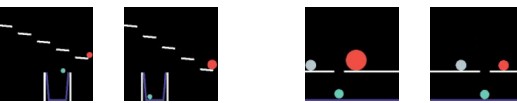

Figure 8: **Pairs of PHYRE scenes in the same template** with similar objects in the background and differences in objects' positions.

**OBJ3D** We collect the OBJ3D dataset from the official GitHub repository[4].

**CLEVRER** We directly download the CLEVRER dataset from the official website[5].

**PHYRE** We use the PHYRE-1B version, which sets the number of red balls to 1. The PHYRE dataset is generated based on the instructions provided on the official GitHub page[6].

**Physion** We directly download the Physion dataset from the official GitHub page[7]

## C   Implementation Details

### C.1   Loss Functions

The following is the training objective that follows the prior work [44].

We employ slot reconstruction loss to train the rolled out future frames prediction, as described in Equation 5 where $n$ is the slot index, $s_{T+k}^n$ is the predicted rollout slot, and $s_{T+k}^{*n}$ is the pre-trained slot (that is used as the target slot).

$$\mathcal{L}_S = \frac{1}{K \cdot N} \sum_{k=1}^{K} \sum_{n=1}^{N} \|s_{T+k}^n - s_{T+k}^{*n}\|^2 \tag{5}$$

Experiments using SAVi as the object-centric model also use the image reconstruction loss, as described in Equation 6 where $f_{dec}$ is a frozen decoder, and $x_{T+k}$ is the ground truth image.

---

[4]github.com/zhixuan-lin/G-SWM#datasets

[5]clevrer.csail.mit.edu/

[6]github.com/facebookresearch/phyre

[7]github.com/cogtoolslab/physics-benchmarking-neurips2021#downloading-the-physion-dataset

Experiments using STEVE as the object-centric model can still employ the image reconstruction loss; however, we do not conduct such experiments with image reconstruction loss due to the dramatically extended training time. In PHYRE, we do not utilize the image reconstruction loss $\mathcal{L}_I$ due to the large image size that could affect the training time and the simplicity of PHYRE's objects compared to other environments in this work.

$$\mathcal{L}_I = \frac{1}{K} \sum_{k=1}^{K} \| f_{dec}(s_{T+k}) - x_{T+k} \|^2 \tag{6}$$

The overall objective function is the weighted sum of the above losses, as presented in Equation 7.

$$\mathcal{L} = \mathcal{L}_S + \lambda \mathcal{L}_I \tag{7}$$

Note that when using slot and image reconstruction losses as presented in Equation 5 and Equation 6, the model will fail in the end-to-end training since the overall objective function is constrained to pre-trained slots ($s_{T+k}^{*n}$) and well-trained decoder ($f_{dec}(\cdot)$). Motivated by this observation, in Appendix E, we provide the end-to-end training objective to compare RSM and other approaches' abilities in objects' dynamics modeling from scratch.

## C.2 Models Architecture

The distinction among methods is in the process of predicting the next state based on the input of past states. In this section, we provide a detailed description of how both the baselines and the proposed RSM approach the task of the next state prediction. Specifically, we examine two key aspects: (1) how slots communicate with each other and (2) how the predictions are computed, as revealed through the design specifics of each method.

### C.2.1 Baselines

**SlotFormer** [44] SlotFormer consists of 3 main parts: (1) the Multi-Layer Perceptron (MLP) input projection layer, (2) a Transformer architecture layer, and (3) the MLP output projection layer. First, the unrolled past states (the sequence of $\tau \times N$ slots) are passed through the input projection layer before being processed by the Transformer model. Afterward, the output projection produces the next state from the Transformer's output. Through this process, slots densely communicate with each other in all three parts of SlotFormer. In addition, the entire next state of slots is directly generated by the models. In this work, we utilize the publicly available implementation[8] of SlotFormer.

**SwitchFormer** from Switch Transformer [14]. The Switch Transformer is a variant of the Transformer architecture designed to improve efficiency and scalability. It achieves this through the usage of dynamic routing and adaptive computation. This architecture employs a "switch" module that intelligently routes tokens to different layers based on their content.
In this work, we create SwithFormer that integrates the Switch Transformer implementation[9] into the SlotFormer codebase and replace the vanilla Transformer by Switch Transformer. In this way, SwitchFormer follows the same strategy as in SlotFormer, which conducts the dense communication among slots and directly predicts the entire next state of slots.

**NPS** [17] NPS is a framework that combines neural networks and production systems for object modeling that integrates neural networks into the production system. In traditional production systems, rules are used to represent knowledge and guide the system's behavior. In the case of NPS, they conduct a set of rules to handle the pair-wise interaction of slots. The two slots involved in an interaction, which are called the *primary* and *contextual* slots, are selected through attention mechanisms. In addition, the official design of NPS for object dynamics' modeling, inspired by Kipf et al. [24], predicts the changes of the *primary* slot within a time step instead of the entire slots. Afterward, the predicted next state of slots is the sum of the current state and the predicted slots changes.
In this work, we integrate the official NPS[10] to the SlotFormer's codebase for consistency in the training pipeline and sharing the pre-trained object-centric model. Besides, NPS includes an MLP

---

[8]github.com/pairlab/SlotFormer

[9]nn.labml.ai/transformers/switch/index.html

[10]github.com/anirudh9119/neural_production_systems

slot encoder that requires a fixed input size. However, the temporal window size in PHYRE increases from 1 to 6, leading to the difference in the input size of the starting and later steps. Therefore, at the beginning of the rollout prediction process of PHYRE, we duplicate the burn-in frame to have a fixed six steps window size along the rollout process.

### C.2.2   RSM

We design RSM as a framework for dynamics modeling with a relaxed inductive bias in the communication density of slots that enables a subset of slots involved in communication, based on a particular context through the CCI. RSM consists of three main elements as described in Section 2.1: (1) the multi-head self-attention that computes the CCI, (2) the $\psi(\cdot)$ that estimates the suitable mechanism, and (3) a list of mechanisms. In terms of the multi-head self-attention, We employ a 4-head architecture for multi-head self-attention, where the hidden size of the Feed-forward Networks is set to $2 \times d_s$. We design $\psi(\cdot)$ as MLP with the number of hidden layers being tuned using the validation set (See Appendix D.3). Similarly, each individual mechanism is designed as MLP layers with a tuned number of hidden layers. All mechanisms share the same architecture but have separate weights. In addition, the total number of parameters when considering all mechanisms is constrained to be the same across when the number of mechanisms varies, meaning that as the number of allocated mechanisms increases, the number of parameters per each mechanism decreases (further investigated in Appendix D.3). Like NPS, the mechanism predicts the changes in a slot within two consecutive steps instead of the entire slot. Last but not least, we (and NPS) omit the input and output projection layers as SlotFormer and SwitchFormer.

### C.2.3   Downstream tasks

**CLEVRER VQA** Inheriting from the baseline of Wu et al. [44], we employ Aloe as the VQA model that concatenates the predicted rollout slots and the processed question (represented as language tokens) before passing them through a stack of Aloe Transformer encoder to predict the answers.

**Physion VQA** In the VQA task of Physion, the objective is to determine whether the red object will come into contact with the yellow object once the dynamics of all the objects have been completed. Since the task does not involve any language processing, we construct an MLP model that takes the rollout slots as input. The MLP processes these slots and produces a binary prediction, indicating whether the red object and the yellow object "touched" or "did not touch".

**PHYRE Readout** In the PHYRE action planning task, an action involves determining the size and position of the red ball. In our approach, we utilize the set of 10,000 predefined actions introduced by [4] and train a readout model to determine if a given action can solve the task. To construct the readout model, we draw inspiration from Wu et al. [44] that design a 2-layer MLP model on top of the encoded states. The readout model takes the predicted rollout states as input. To process these states, we employ an encoder, which differs depending on the specific model variant used. In SlotFormer, a Transformer is used, while in SwitchFormer, a Switch Transformer is employed. An MLP is operated as the encoder in the NPS model, and a Multi-head Self-Attention mechanism is utilized in the RSM model. Once the encoder has processed the states, the classifier generates a binary output indicating whether the task has been solved. This output serves as an inference for the task's solvability.

## D   Further Discussion on Experiment Results

### D.1   Rollout Future Prediction

Figure 9 depicts the rollout frames generated by RSM alongside the baselines. Notably, RSM excels in producing robust future frames that accurately capture the dynamics of objects while maintaining visual fidelity. Nevertheless, we have encountered a challenge in generating objects with sharpness within the CLEVRER dataset. SlotFormer marks partially incorrect object's dynamics in this case.

### D.2   Discussion on the Action Planning task in PHYRE

We encountered difficulties in reproducing the action planning results of SlotFormer, even when using their provided checkpoints. In Table 3, we report the iid result of 76.4 for SlotFormer, which is 5.6 points lower than the officially reported value. To investigate this issue, we explored potential reasons

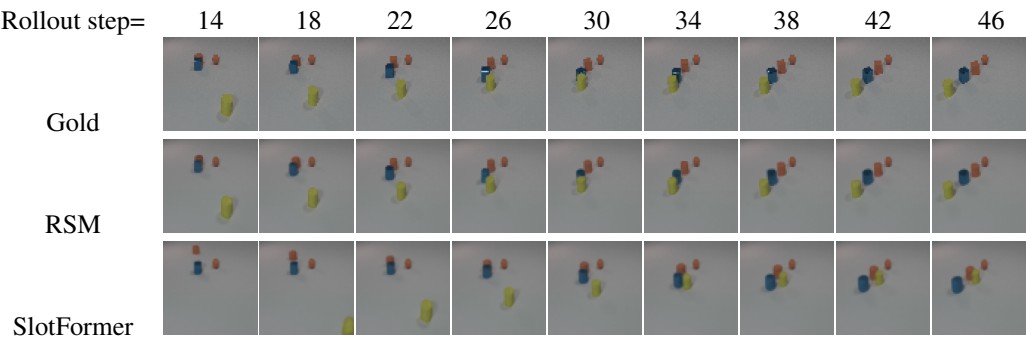

Figure 9: Comparison of rollout frames in CLEVRER.

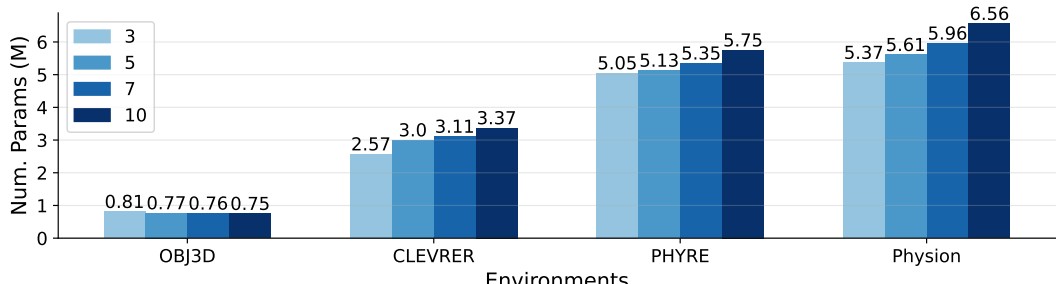

Figure 10: The scaling of RSM's parameters with different amounts of mechanisms.

and identified the following possible factors: (1) Instability of results: The official GitHub page of SlotFormer[11] acknowledges that the results in this specific task may exhibit instability. This suggests that achieving consistent and reproducible results with SlotFormer can be challenging and (2) data discrepancies: The PHYRE dataset, being regenerated rather than downloaded from a common source, introduces variations in the computing configuration. These data collection and processing differences may contribute to the disparities observed between our results and the reported values in Wu et al. [43].

### D.3 Finetuning Results in RSM

In Figure 11, we present the results of hyperparameter fine-tuning on CLEVRER and Physion datasets. This section explores the impact of the number of mechanisms, the expansion of $\psi(\cdot)$ parameters, and the structure of the mechanism models. The term *number of layers* in this analysis refers to the hidden layers within the MLP structure that maps the input dimension to the output dimension in the respective models.

In terms of the number of mechanisms, we have observed that having either a large or a small value for $M$ significantly worsens the results and leads to high variance. However, we have also discovered that the tasks can be solved with relatively few mechanisms, even when dealing with diverse object movements. To validate the design principle stated in Appendix C.2.2, which proposes an inverse relationship between the number of the mechanism's parameters and the number of mechanisms, we conducted experiments denoted as *10\**, which replicates mechanisms to achieve a total of 10 mechanisms without reducing the number of parameters, as compared to the best configuration (the first boxes). Our findings indicate that the replicated configuration (10\*) achieves a slightly lower score than the best configuration, and the additional mechanisms are not selected by the $\psi(\cdot)$ models.

In terms of the number of layers in $\psi(\cdot)$, the challenge is to map a $2 \times d_s$ vector to a compact vector of size $M$. Our findings suggest that using 1 or 2 hidden layers yields favorable results in terms of achieving a high score. However, we find a complication in identifying consistent patterns for fine-tuning $\psi(\cdot)$ across different datasets.

---

[11]github.com/pairlab/SlotFormer

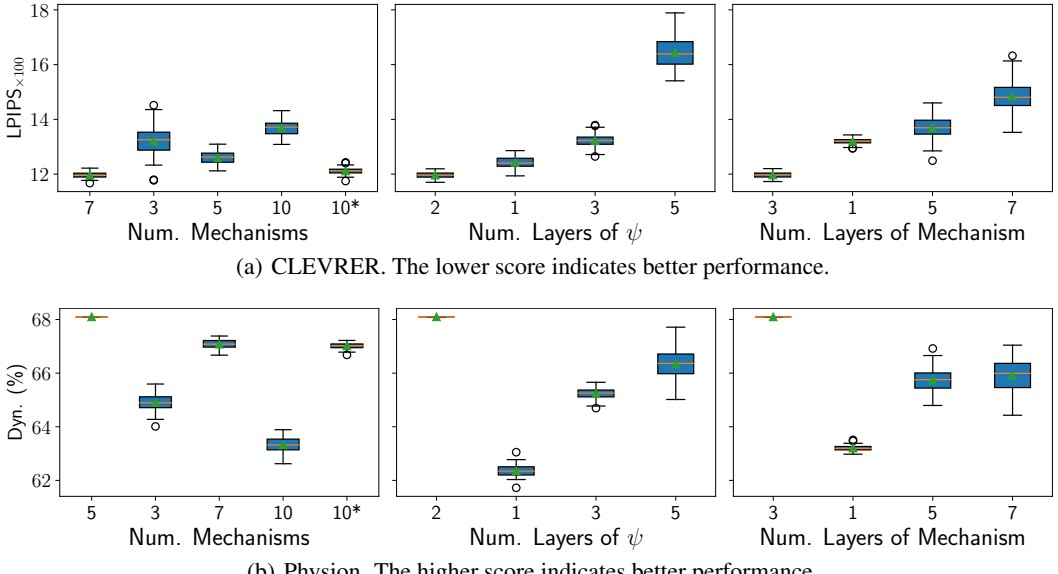

(a) CLEVRER. The lower score indicates better performance.

(b) Physion. The higher score indicates better performance.

Figure 11: **The finetuning results in CLEVRER and Physion.** The first box of each subplot is the configuration that achieves the best performance, whereas other boxes are performance that has a different number of mechanisms, layers of the $\psi$ model, or layers of each mechanism from the best setting. The plotted values are the mean and standard deviation over 5 different runs corresponding to each configuration. See text for more details.

Lastly, when considering the mechanism structure, we have noticed that increasing the number of parameters allows a single mechanism to achieve a moderately decent score, although not a high score. Consequently, the $\psi(\cdot)$ model tends to choose only one mechanism for all cases, resulting in a lower score when assigning 5 or 7 layers to the mechanism.

# E  Experiments on End-to-end Training Pipeline

We propose an additional experiment to verify the methods' ability to model objects' dynamics from scratch, additionally, in action-conditioned environments. We establish an end-to-end training pipeline that comprehensively assesses the models' effectiveness in the entire process of extracting slots from frames, handling the objects' action-conditioned dynamics, and finally decoding slots back into frames.

RSM generally demonstrates a superior ability to model objects' dynamics and produce meaningful slots compared to the baselines.

## E.1  Experiment Setup

**Environment:** This dataset consists of objects arranged in a $5 \times 5$ grid. At each time step, a single frame and an action are provided. The action specifies one object and one manipulation from the set of UP, RIGHT, DOWN, and LEFT. The challenge of this task is to determine the feasibility of the given action and predict the resulting frame. For example, an action of moving an object to the LEFT is executable if no objects are obstructing the left side of the target object. Otherwise, the frame remains unchanged.

**Encoder and Decoder architecture** In this experiment, we follow the encoder proposed by Kipf et al. [24] that contains a simple CNN-based *Object Extractor* to extract $N$ feature maps from the input frame that is followed by an MLP-based *Object Encoder* to encode feature maps to vector representations of objects, *i.e.* the object slots. Afterward, we exploit a decoder architecture for slot-based visual prediction, consisting of $N$ slot decoders with separate weights. Each slot is decoded into an RGB reconstruction, and the final frame reconstruction is obtained by summing the reconstructions of all slots.

**Training Objective.** This experiment follows the Contrastive loss setup as Kipf et al. [24] that uses the prediction result of the transition model to form the positive hypothesis, whereas, sampling random input states in the same batch forms the opposing hypothesis. The target slots, $s_{t+1}^{*1:N}$, are obtained by passing the target next frame through the Encoder, $s_{t+1}^{1:N}$ denotes the predicted slots, and $\tilde{s}_t^{1:N}$ indicates the random slots in the training batch.

$$H = \mathsf{MSE}(s_{t+1}^{1:N}, s_{t+1}^{*1:N}), \quad \tilde{H} = \mathsf{MSE}(\tilde{s}_t^{1:N}, s_{t+1}^{*1:N})$$
$$Contrastive\ Loss : \mathcal{L} = H + \mathsf{max}(0, 1 - \tilde{H}) \tag{8}$$

We consider the sum of two BCE loss terms in training Decoder, $\mathcal{L}_1$ and $\mathcal{L}_2$. $\mathcal{L}_1$ is applied on $x_t'$, obtained by passing $s_t^{1:N}$ through Decoder, which is expected to be close to the input frame $x_t$. $\mathcal{L}_2$ is applied on $x_{t+1}'$, obtained by passing $s_{t+1}^{1:N}$ through Decoder to achieve the prediction of next frames reconstruction, which is desired to be close to the target next frame $x_{t+1}$.

$$x_t' = \mathsf{Decoder}(s_t^{1:N}), \quad x_{t+1}'^{1:N} = \mathsf{Decoder}(s_{t+1}'^{1:N})$$
$$\mathcal{L}_1 = \mathsf{BCE}(x_t', x_t), \quad \mathcal{L}_2 = \mathsf{BCE}(x_{t+1}', x_{t+1}) \tag{9}$$
$$BCE\ Loss : \mathcal{L}_{Decoder} = \mathcal{L}_1 + \mathcal{L}_2$$

## E.2 Experimental and Analytical Results

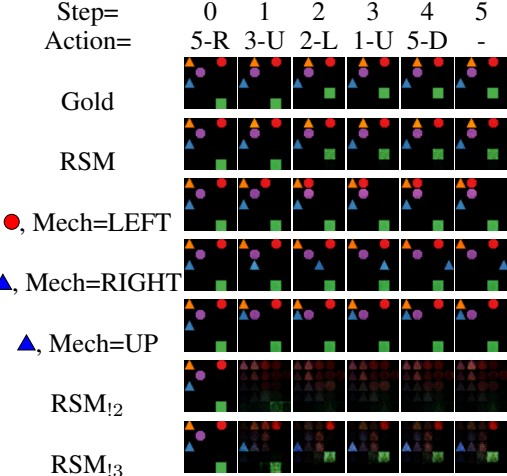

Figure 12: Observation of mechanisms assignment and performance in 5 rollout steps in 2D Shapes. In the last three rows, we present the reconstruction by only changing 1 slot with 1 mechanism applied to that slot over 5 steps, while all other slots are untouched.

**Disentangling objects transition dynamics to mechanisms.** In Figure 12, we study the role of each mechanism in RSM. The analysis shows that RSM produces a reasonable reconstruction compared to the ground-truth frame and encourages the mechanisms to distinguish themselves in their roles. In this sample, we can infer the 5 slots corresponding to: **1**: red circle, **2**: blue triangle, **3**: green square, **4**: purple circle, and **5**: yellow triangle. Likewise, we observe the mechanism assignment in each step and list the role of each mechanism (5 mechanisms in this experiment) specialized as the following: Mechanism **1**: RIGHT, Mechanism **2**: UP, Mechanism **3**: LEFT, Mechanism **4**: *DO NOT MOVE*, and Mechanism **5**: DOWN.

Looking deeper into the reconstruction results, RSM takes advantage of the CCI to assign a suitable mechanism for each slot in all scenarios, considers the particular situation, and reacts accordingly to the same action given the specifics of each context. For instance, we observe that with the same action UP given in step 1 on the green rectangle and step 3 on the red round, RSM recognizes the situations in which the object is allowed to move or is blocked by the upper wall, respectively, then applies the movement at step 1 while not modifying objects at step 3 and generates the correct reconstruction in both cases. A similar example is shown in the row corresponding to mechanism 2 at step $3 \rightarrow 4$ when the green object does not move UP and remains at the same position since the red object blocks it.

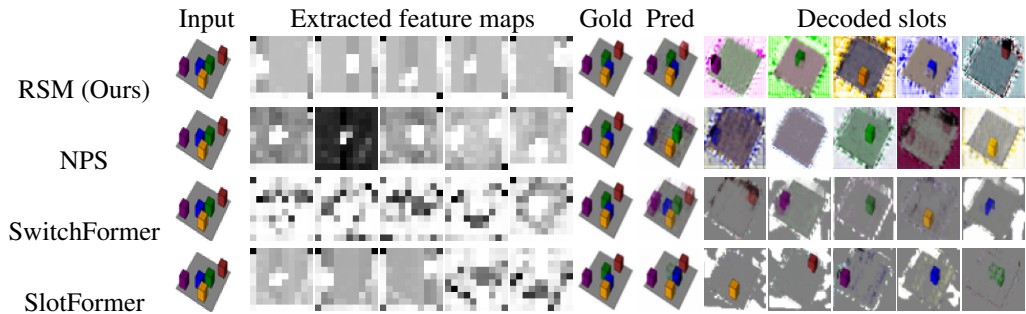

Figure 13: Comparison of extracted feature maps from a scene and reconstructions in 3D Cubes. RSM deals with object slots better than baseline in both slot extraction and slot decoding phases.

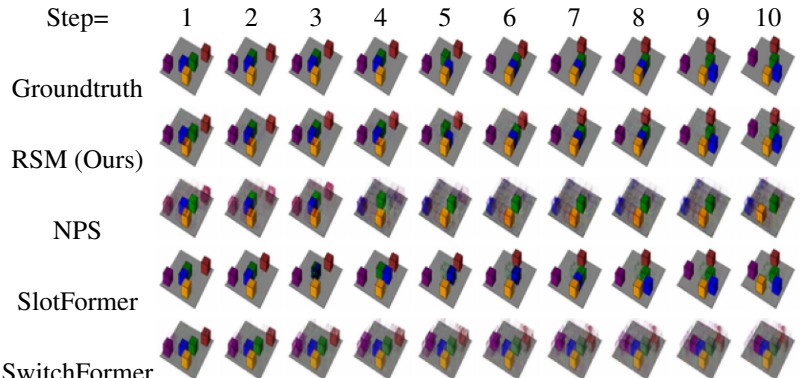

Figure 14: Reconstruction comparison on 3D Cubes dataset

**The ability to decompose a frame into slots.** We analyze RSM's slot-centric representation learning ability. In Figure 13, we illustrate a comparison of the extracted feature maps with a size of $10 \times 10$, which are constructed by the Encoder model and the reconstructed slots and the frame with a size of $3 \times 50 \times 50$, acquired by the SlotDecoder models that receive the input as the predicted next state. We find that RSM decomposes the input frame into separated slots and keeps each object in the same slot until the decoding phase. In contrast, the baselines do not capture all objects but produce noisy feature maps (SlotFormer and SwitchFormer), or put the same object in two slots and identify two objects in another slot (NPS).

We observe that combining the CCI with the sequential updates that encourage slots to observe the modification of each other benefits from recognizing the overlap of objects. More specifically, RSM only generates a part of the object in case that object is covered by another one (*e.g.*, the green and blue cubes in slot 2 and slot 4 are partially covered by the yellow cube). On the other hand, the baselines overlook the condition of executability of action and generate overlapped objects in some cases (*e.g.*, the green, blue, and yellow cubes in SlotFotmer overlapping one another). Lastly, SwitchFormer produces blurry objects and incorrectly predicts objects' dynamics.

**Reconstruction in 3D Cubes**. One of the challenges of generating reconstructions in 3D Cubes is to recognize the visibility order of objects. Figure 14 illustrates RSM's strength in communication among slots to obtain the order information, as well as its ability to generate the proper movement of slots and produce an accurate reconstruction compared to the ground truth. In contrast, SlotFotmer misses that kind of information from the beginning steps and renders the blue and green objects inside each other. Besides, other methods lose the information about some objects and produce inaccurate reconstructions at the end, witnessing a huge gap from the preceding steps up until step 10.

# F  Limitations and Future Works

While RSM has demonstrated robustness for modeling objects' dynamics in various tasks in both iid and OOD settings, there is a lot of room to expand this work to overcome the following limitations:

1. Sensitivity to Hyperparameters: RSM requires tuning the number and size of mechanisms. Future research could explore automated methods for determining optimal values and enhancing RSM's adaptability across tasks and scenarios.

2. Time complexity: The sequential update of slots can become computationally expensive in larger systems with many slots (e.g., exceeding 10,000). Although this problem does not affect our work (see Appendix A), future research should also explore parallelization techniques to improve computational efficiency by assigning mechanisms and predicting the next states simultaneously for large-scale systems.

3. Environments in this study are all observable. Future work should explore a wider range of observable and unobservable environments to gather deeper insights into the working mechanism of RSM in such domains.

