# OpenReview forum: "Reusable Slotwise Mechanisms"
_NeurIPS.cc/2023/Conference — NeurIPS 2023 poster_

### Official Review · Reviewer_TEgG · 2023-07-04

**Soundness:** 2 fair
**Presentation:** 3 good
**Contribution:** 3 good
**Rating:** 7
**Confidence:** 4

**Summary:**

This paper presents an architecture (Reusable Slotwise Mechanism or RSM)  for modelling the temporal dynamics of objects based on a slot-represent.
The main idea is to extract a global context (Central Contextual Information or CCI)  from the past frames and use this to stochastically select one of a small discrete set of (learned) mechanisms for each slot and time-step. The mechanism computes the update of a slot,  based on its current representation as well as the CCI. The slots are updated sequentially. It is shown anecdotally that they specialize to particular object behaviors/dynamics (such as idle, free-fall and collision) and are reused across slots and time.
RSM is compared to several strong baseline methods on several tasks including dynamics modelling, question answering, and task planning. It demonstrates significant improvements over the baselines with a comparable amount of parameters and compute.


**Strengths:**

* The studied problem of modelling temporal dynamics from raw video is an important one, and there still is a lot of headroom.
* The idea behind RSM is simple and well motivated.
* RSM clearly improves upon the baselines and produces SOTA results on several relevant benchmarks. It thus presents an important step towards unsupervised learning of dynamics models from video.
* The presentation is clear and the paper + supplementary contain all relevant information for reproducing the experiments.

**Weaknesses:**

To evaluate the different parts of the RSM architecture, the paper presents 4 ablations in section 4.5. Unfortunately the only quantitative results in Fig 5. the present fell out of the main paper into the appendix.

But if I understand correctly there is a more severe problem: it seems from the text that ablations RSM!2, RSM!3 and RSMk  are **inference time** ablations. That means during training the model had access to CCI in steps 2 and 3 and has learned to choose a mechanism. It is thus not at all surprising that changing any part of that during inference will deteriorate the performance. Thus, all that these ablations demonstrate is that the model has not actively learned to ignore the CCI and the mechanisms.
It would be much more informative to see how the model performs if these modifications are also applied during training. That could for example help answer the much more interesting question "How much does having access to CCI during step 2 help the model choose the right mechanism". For this very reason ablation 4 is much more informative.
In my opinion the such training-time ablations are necessary to properly support the claims about the role and importance of CCI and the mechanisms.



Other minor weaknesses:
* the differences between RSM and the baselines could be made clearer. There is some information in Sec 3.2 but it doesn't clarify what exactly is different.
* The citations especially in the introduction and related work seem a bit biased with many citations to Goyal et al while leaving out other relevant and influential work in the area such as:
  - Eslami, S. M., Heess, N. & Weber, T. Attend, infer, repeat: Fast scene understanding with generative models. Adv. Neural Inf. Process. Syst. (2016)
  - Burgess, C. P. et al. MONet: Unsupervised Scene Decomposition and Representation. arXiv [cs.CV] (2019)
  - Greff, K., Kaufman, R. L. & Kabra, R. Multi-object representation learning with iterative variational inference. International (2019)
  - Battaglia, P. W. et al. Relational inductive biases, deep learning, and graph networks. arXiv [cs.LG] (2018)
  - Greff, K., van Steenkiste, S. & Schmidhuber, J. On the Binding Problem in Artificial Neural Networks. arXiv [cs.NE] (2020)
  - Schölkopf, B. et al. Toward Causal Representation Learning. Proc. IEEE 109, 612–634 (2021)

* Sec 2.2 feels a bit repetetive with the enumeration of the three main components of RSM and the list of four steps in RSM mostly explaining the same things.


**Questions:**

* My main request would be training-time ablations for $RSM_{!2}$ and $RSM_{!3}$ with quantitative results, since they would help buttress the claims about the importance of the CCI.
* Similarly, the paper claims in various places that the CCI has advantages over pairwise interactions. Is there evidence from a fair comparison of the two? Communication is pairwise in NPS, but is that the only relevant difference?
* How big is $d_{cci}$? The paper claims that it acts as a bottleneck, yet the CCI is recomputed for every slot from a self-attention among all previous slots. So unless $d_{cci}$ is at least smaller than $d_s$ I do not see in what sense it constitutes a bottleneck.
* Are all mechanisms equally important? I suspect that main advantage of RSM might be the ability to put objects firmly in an idle mode.

**Limitations:**

The paper discusses some limitations in the supplementary. I see no necessity for a discussion of negative societal impact.

---

> ### Author Rebuttal · Authors · 2023-08-10
>
> We deeply appreciate the time and effort you invested in evaluating our work.
>
> We are glad to see `simple and well motivated`,  `clearly improves `,  and `presentation is clear `  in your reviews.
>
> ---
> > **training-time ablations for $RSM_{!2}$  and  $RSM_{!3}$  with quantitative results.**
>
> We   agree that the training-time ablations for $RSM_{!2}$  and  $RSM_{!3}$  are helpful.
>
> First, we would like to repeat the   purpose behind  the ablations $RSM_{!2}$ and $RSM_{!3}$. CCI is   in steps 2 and 3 in a concatenation, a   question arises: Is the CCI really influence     these steps, or is the greater weight being placed on the other component of the concatenation? Therefore,  we mask out the CCI in these steps to verify.
>
> Next, we present the results of our training-time ablations on OBJ3D across 5 random seeds. Ablations with a `'`  for  training-time, while those without   `'`  for   inference-time.
> | Method |  LPIPS$_{\times 100}\downarrow$|
> |--|--|
> | **$RSM$** | **7.88**±0.1 |
> |**$RSM_{!2}$** | 15.32±0.4 |
> |**$RSM_{!3}$** | 9.36±0.2 |
> |**$RSM_{!2}'$** | 17.3±0.9 |
> |**$RSM_{!3}'$** | 9.70±0.2 |
>
> We could see that:
>  -  **$RSM$** achieves the best performance over the ablations.
>  - **$RSM_{!2} '$** experiences challenges when using   one slot as input for mechanism selection, then tends to favor a single mechanism after a few epochs consistently. This stresses the vital role of the CCI  in   information diffusion among slots during mechanism selection.
>   - **$RSM_{!3}'$** outperforms $RSM_{!2}'$ but lags behind RSM by a large margin.
> 	  * Using a single slot as mechanism input is not entirely detrimental if CCI aids in recognizing mechanism roles. Therefore, it does not produce a really bad performance.
> 	  * Nonetheless,  Section 4.5 underscores that CCI also holds contextually   data guiding slot movements (e.g., left, right, or some angle). Therefore, the mechanism still needs the CCI  for a more accurate prediction.
> 	  * Furthermore, the mechanism selection process generates a probability   for M mechanisms' indices, which must convey the entire system context due to its limited size. That's why the CCI   in the mechanism's input helps deliver system context.
>
> These findings show the key role of CCI in   slots' communication.
>
> ---
> >  **Communication is pairwise in NPS, but is that the only relevant difference?**
>
> Yes, the pairwise communication in NPS is the only relevant difference. The rest is conceptually equivalent for RSM and NPS, such as the training pipeline, object-centric models, and configurations.
>
> ---
> > **How big is  $d_{cci}$?** The paper claims it acts as a bottleneck, yet the CCI is recomputed for every slot from a self-attention among all previous slots.
>
> Thank you for your question. We choose $d_{cci}=d_s$ in this work.
>
> We call CCI a bottleneck not merely because of its dimension; rather, CCI is summarizing the information from all slots, i.e., $N\times d_s$, in a vector of size $d_{cci}=d_s$. Moreover, across rollout steps, the system communicates and joins to update the CCI. That is what we mean by a bottleneck. It is indeed recomputed from a self-attention among all previous slots, but it constantly summarizes the whole state of all earlier slots as they change in a fixed-size vector, squeezing the information.
>
> We hope this helps clarify what we denote as a bottleneck.
>
> ---
> > **Are all mechanisms equally important?**
>
> Thank you for  noticing this vital point.
>
> All mechanisms have an equal chance of getting selected at the beginning, as there are no pre-defined roles. They share the same architecture and are  initialized randomly.
>
> Through training, it gains the ability to decompose the transitions into separate mechanisms with varying importance. Certain environments may require specific actions, we expect RSM to **autonomously  recognize** these characteristics and adjust the frequency of mechanisms accordingly.
>
> This table is the ratio of mechanisms selected in test time over 3 datasets with a single seed.
>
> | Mechanisms  | OBJ3D | CLEVRER|  PHYRE|
> |--|--|--|-- |
> | **1** | 0.435 |  0.037| 0.163
> |**2** | 0.107 | 0.370  |  0.057
> |**3** | 0.045 | 0.076 |   0.144
> |**4** | 0.096 | 0.129 |   0.461
> | **5** | 0.205 | 0.212  |  0.173
> | **6** | 0.044   |   0.126|        -|
> | **7** | 0.064 | 0.047 |   -
>
> The 7 mechanisms in OBJ3D can be identified as: Idle (1), Moving fast (2), Moving and spinning (3), Collision (4), Redundant   (5,6), and Moving slow (7).
>
> The 7 mechanisms in CLEVRER can be identified as: Redundant   (1,6), Idle (2), Collision (3), Moving slow (4), Moving fast (5), Moving and spinning (7).
>
> The 5 mechanisms in PHYRE can be identified as: Redundant    (1), Collision (2), Moving left or right (3), Idle (4), and Falling (5).
>
> Please refer to **Section "2. Mechanisms assignment in OBJ3D and CLEVRER" in the Global Rebuttal** for discussing the **Redundant Mechanisms**.
>
> > **I suspect that main advantage of RSM might be the ability to put objects firmly in an idle mode.**
>
> Yes, we think recognizing the idle status of objects is an important ability. However, 2 points are also necessary for a successful prediction: (1) the accurate assignment of all other mechanisms  and (2) the accurate prediction of the `Idle` mechanism for a smooth communication of the slots.
>
>
> ---
> > the only quantitative results in Fig 5. the present fell out of the main paper into the appendix.
>
> We are so sorry for the confusion on Figure 5's position.  We have moved it back to the main text in our latest version.
>
> > The differences between RSM and the baselines could be made clearer.
> >
> > The citations  seem a bit biased with many citations to Goyal et al
> >
> > Sec 2.2 feels a bit repetetive with the enumeration of the three main components of RSM and the list of four steps.
>
> We truly appreciate your suggestions. We will certainly fix these points in the final version and make  adjustments for a fairer presentation of the prior work.

---

> > ### Comment · Reviewer_TEgG · 2023-08-18
> > **Answer to Rebuttal**
> >
> > I would like to thank the authors for their comprehensive rebuttal. I appreciate the additional experiments run. It is surprising to me that the training time ablations generally perform worse than the corresponding test-time ablations. I would have expected the model to perform better when training and test settings are identical.
> >
> > Overall I think the additional results and clarifications do improve the paper, and I reaffirm my recommendation of a clear accept.

---

> > > ### Author Response · Authors · 2023-08-19
> > >
> > > Thank you for taking a careful look at our rebuttal. We're glad that our rebuttal alleviated the reviewer's concerns.

---

### Official Review · Reviewer_8CU7 · 2023-07-04

**Soundness:** 2 fair
**Presentation:** 2 fair
**Contribution:** 3 good
**Rating:** 6
**Confidence:** 3

**Summary:**

This paper presents a study on future prediction using object-centric representations and disentangled and reusable mechanisms (RSM) that govern the interactions between objects. The authors' key contribution lies in the proposal of Central Contextual Information (CCI) representation, which captures the interactions between objects observed in the past. The CCI is encoded using multi-head attention layers with all object-centric representations from previous observations and a bottleneck projection layer. It aids the model in selecting a single mechanism to predict object transitions from the current state to the next state. Experimental results demonstrate that the introduced RSM outperforms several competing baselines in tasks such as future frame prediction, visual question-answering, and action planning in relatively simple environments.

**Strengths:**

+) The motivation behind the proposed CCI is clearly explained, with the authors using multi-head attention layers to capture interactions between objects in past observations. The subsequent bottlenecking of the attention layers’ output into the CCI enables the model to learn a concise representation that encapsulates meaningful information about object interactions.

+) The selection process for predicting state transitions is intriguing and surprisingly effective. The use of Gumbel-softmax (assuming it is Gumbel-softmax instead of Gumbel-max as mentioned in the manuscript) allows the model to decompose the potential mechanisms governing state changes.

+) The visualizations in Figures 4 and 5 provide compelling evidence of the model's ability to decompose mechanisms into discrete and reusable components. Figure 4 demonstrates the model's capability in decomposing mechanisms, while Figure 5 showcases how the CCI encodes essential information for determining outcomes following object interactions.

+) The experimental results support the efficacy of the proposed RSM, as it outperforms several competing baselines, including SlotFormer, SwitchFormer, and NPS, across various tasks such as future frame prediction, visual question-answering (which relies on future visual states), and action planning.

**Weaknesses:**

-) The assumption that there is only one mechanism responsible for the state change of an object may not always hold true. Factors such as the object's original momentum and collisions with other objects can influence its behavior. Therefore, the use of Gumbel-softmax (again, assuming it is Gumbel-softmax instead of Gumbel-max as mentioned in the manuscript) in Equation (2) might impose overly strong restrictions on modeling interaction mechanisms.

-) There appears to be a misinterpretation regarding the mention of the Gumbel-max layer in Step 2. The Gumbel-max layer, without the Straight-Through (ST) Gumbel Trick, is not differentiable and is primarily used to enable the categorical sampling using unnormalized log probabilities by the reparameterization trick. The references cited at Line 122 also pertain to Gumbel-Softmax, not Gumbel-max. For more details, please refer to the PyTorch implementations of the Gumbel-Softmax layer: https://pytorch.org/docs/stable/generated/torch.nn.functional.gumbel_softmax.html#torch.nn.functional.gumbel_softmax or the TensorFlow API: https://www.tensorflow.org/probability/api_docs/python/tfp/distributions/RelaxedOneHotCategorical.

-) There is a discrepancy at Line 269 where the authors mention the visualization of disentangled mechanisms in Figure 5. However, Figure 5 is not found in the main submitted manuscript but in the supplementary material (Page 13). The mention of Figure 5 at Line 269 implies it should be present in the main paper, potentially violating the Formatting instructions.

-) While the proposed RSM outperforms competing baselines on various datasets and tasks, it is worth noting that the experiments are conducted in relatively unrealistic environments with toy objects and simple interactions. It would be beneficial to demonstrate the effectiveness of the proposed approach (object-centric representations + RSM) on more realistic datasets such as MOVi-B, C, D, E, and F.

**Questions:**

o) The usage of the non-differentiable Gumbel-max layer needs clarification. If it is indeed used, how are the gradients backpropagated to the layer $\psi$? If not, is it a typo?

o) In Equ. (3), if only one mechanism is selected, why does the formulation still require a sum operation to combine all other transitions predicted by other mechanisms?

o) Have the authors experimented with using the Softmax operation in Equ. (2) instead of the hard selection provided by the Gumbel-softmax?

o) Can the authors demonstrate the scalability of the visualization of mechanism assignments/decompositions with more scenarios? For example, can they provide additional visualizations of mechanism decompositions with different initial states/transitions on different datasets such as OBJ3D or CLEVRER?

o) In Line 101, when mentioning the projection $\phi$, is it referring to a linear layer or an MLP (multi-layer perceptron)?

**Limitations:**

-) As mentioned in the Weaknesses, the experiments conducted in this study are limited to relatively unrealistic environments with toy objects and simple interactions. As a result, it remains uncertain whether the proposed approach would maintain its effectiveness in a more realistic environment or when applied to real-world data.

---

> ### Author Rebuttal · Authors · 2023-08-10
>
> Thank you for your valuable reviews. We deeply appreciate the time and effort you invested in evaluating our work.
>
> We are happy to see `clearly explained motivation`, `intriguing and surprisingly effective`, and `impressive`  in your reviews.
>
> ---
>
> > 1. **The usage of the non-differentiable Gumbel-max layer needs clarification.**
> >
>
> We apologize for any confusion regarding using the Gumbel-Softmax in our paper.
> We would like to clarify the matter as follows.
>
> First of all, we agree that we should have mentioned the "Straight-Through Gumbel Trick" in our paper. That is exactly what we are using to backpropagate gradients in training when sampling from a categorical distribution. We employ `F.gumbel_softmax()` with the `hard=True` option (where F refers to `torch.nn.functional`, coherent with your referenced implementation of Gumble-softmax). This enables hard selection from the categorical distribution during the forward pass, and backpropagates the gradients using the straight-through trick in the backward pass.
>
> Once again, we are sorry for the confusion. In light of your feedback, we have made the necessary adjustments to our paper to explicitly include the "Straight-Through Gumbel Trick" terminology, clarifying our use of Gumbel-Softmax.
>
>
>
>
>
> ---
> > 2. **In Equ. (3), if only one mechanism is selected, why does the formulation still require a sum operation to combine all other transitions predicted by other mechanisms?**
>
> Thank you for your question. Equ. (3) just presents a notation convention. The summation has only one non-zero term, and we wrote the equation as such for completeness in case one decides to use a softmax instead of our hard selection. But again, the summation in Equ. (3) contains only one non-zero element. We're sorry if this has confused; we would change it for the final version to avoid any confusion.
>
> ---
> > 3. **Have the authors experimented with using the Softmax operation in Equ. (2) instead of the hard selection provided by the Gumbel-softmax?**
>
> Thank you for your question.
> Our work has intentionally refrained from conducting experiments with the Softmax operation (`F.softmax()`  or `F.gumbel_softmax(hard=False)`)  for a specific rationale. Our primary objective is to achieve the disentanglement of mechanisms, enabling them to independently and autonomously address distinct tasks. In line with this, our design philosophy emphasizes the independence of mechanisms, thereby avoiding significant information sharing among mechanisms. As such, the Softmax operation is not aligned with our adopted direction and causes information leakage among the slots.
>
> Nevertheless, we acknowledge the merit of incorporating a comparative perspective. In this regard, we have compiled a comparison between the performance of RSM when utilizing the Gumbel-Softmax with hard selection and Softmax operations during the training phase. The following table offers an overview of this comparison:
>
> | Method | OBJ3D (LPIPS$_{\times 100 } \downarrow$) |  PHYRE-fold0 (AUCCESS $\uparrow$)|
> |--|--|--|
> | **RSM with Gumbel-softmax and hard selection** | **7.88** ± 0.1 |**82.14** ± 0.2|
> |**RSM with Softmax** | 9.65 ± 0.2 | 74.31 ± 0.5 |
>
> According to the table, we could see that RSM with Gumble-softmax and hard selection produces much better performance than one with Softmax, indicating the importance of learning independent mechanisms in RSM.
>
> ---
> > 4. **Can the authors demonstrate the scalability of the visualization of mechanism assignments/decompositions with more scenarios?  such as OBJ3D or CLEVRER?**
>
> Sure. **Please refer to Figures 2 and 3  in the supporting document for visualizations and Section "2. Mechanisms assignment in OBJ3D and CLEVRER" in the Global Rebuttal for the discussion**.
>
>  ---
> > 5. **In Line 101, when mentioning the projection $\phi$, is it referring to a linear layer or an MLP (multi-layer perceptron)?**
>
> It refers to a linear layer mapping  $\mathbb{R}^{d_s} \rightarrow \mathbb{R}^{d_{cci}}$.
>
> ---
>
> > -) **The assumption that there is only one mechanism responsible for the state change of an object may not always hold true.**
>
> Thank you for your insightful comment. This is an important observation, and we are aware of the limitation and have tried ideas to explore how it affects performance. As shown earlier, the softmax selection of the mechanisms does not improve performance. Alternatively, we could employ multiple loops of the sequential slot updates,   allowing multiple mechanisms to alter a slot's state. However, we did not observe any surprising advantages compared to the original RSM, and the results stayed on par with them. However, this observation does not refute the possibility that more realistic scenes might benefit from such modifications or other ways that we haven't tried for having multiple mechanisms change a slot's content. Therefore, we remain keen on exploring the extent of this limitation in more realistic scenarios.
>
> On the other hand,  we observe that mechanisms' behavior can deal with complex scenarios, such as colliding with another object when moving or moving while spinning. More details for these cases are  in the answer to your question 4 above.
>
>
>  ---
> > -) There is a discrepancy at Line 269 where the authors mention the visualization of disentangled mechanisms in Figure 5.
>
> We are so sorry for the confusion on Figure 5's position.
> Figure 5 should definitely be in the main text. We have moved it back to the main text in our latest version.
>
> ---
> > -) **It would be beneficial to demonstrate the effectiveness of the proposed approach (object-centric representations + RSM) on more realistic datasets such as MOVi-B, C, D, E, and F.**
>
> Thank you for your great suggestion. We have experimented on **MOVi-C** and got some initial results.
> **Please refer to   Figure 1 in the supporting document for visualizations and the "1. Additional experiment on the realistic dataset - MOVi-C" section in the Global Rebuttal for the discussion**.

---

> > ### Comment · Reviewer_8CU7 · 2023-08-18
> > **The authors' responses effectively address all my concerns**
> >
> > I would like to thank the authors for providing detailed responses to address my concerns and confusing points, especially regarding the confusion surrounding the gumbel-softmax issue. I'm content with this matter as long as the final version clarifies that the backpropagation is facilitated by the Straight-Through Gumbel Trick. Additionally, I appreciate the authors' clarification on Equation (3) and Line 101.
> >
> > I appreciate the authors for incorporating new experimental results with the selection mechanism utilizing the softmax operation. While it's interesting that this model slightly underperforms compared to the RSM with gumbel-softmax, I agree with the authors' reasoning that "more realistic scenes might benefit from such modifications".
> >
> > Lastly, I would like to thank the authors for introducing new experiments involving the MOVi-C dataset, along with additional visualizations of mechanism assignment in OBJ3D and CLEVRER. After considering other reviewers' comments, I don't have further concerns at this point. Given the authors' commitment to addressing the major points of confusion in the final version, I have increased to raise my score to Weak Accept.

---

> > > ### Author Response · Authors · 2023-08-19
> > >
> > > Thank you for taking a careful look at our rebuttal and raising the score. We're glad that our rebuttal alleviated your concerns.

---

### Official Review · Reviewer_NsYp · 2023-07-07

**Soundness:** 3 good
**Presentation:** 3 good
**Contribution:** 3 good
**Rating:** 6
**Confidence:** 4

**Summary:**

The paper proposed Reusable slot-wise mechanisms wherein the authors introduce "Central Contextual Information" -- a bottleneck that captures the global context and helps in choosing which slot to attend to.
Through experiments on several simple and relatively complex datasets are shown with reconstruction, VQA and planning tasks to show the benefit of the reusable mechanisms.


**Strengths:**

1. I find the introduction a bottleneck CCI to summarize the context of past $T$ frames to be intuitive and novel.

2. The method shows significant improvement in OOD action planning tasks which is impressive!


**Weaknesses:**

3. **CCI**: Have authors tried treating each slot independently as input to the transformer with positional encoding instead of concatenating past $T$ time slots?

4. In Fig 1(c), is should the $t$ in blue circle (drawn adjacent to the CCI bubble) be $s_t$. If so please clarify in the figure as $t + \delta s_t $ in Step 3 is confusing as one represents time and another represents the predicted change in slot representation.

5. **Memory Constraints**: The CCI (Step 1) has a huge memory constraint of $(\tau+1) \times N \times d_s$ which is $7 \times 6 \times 128 = 5376$ for OBJ3D dataset (which has the smallest memory requirements), and $16\times 6 \times 192 = 18432$ which is huge. This appears to be a huge factor given that object-centric models are already memory expensive when it comes to applications such as RL.

6. **OOD in dynamics**: How does RSM behave when it sees an OOD dynamics for a particular object. For example, in the OBJ3D dataset (or any other simplistic dataset with enough variations in objects and their dynamics) if a green object is static throughout during the training, but during testing scenario if the green object moves would RSM reuse the mechanism corresponding to the motion of object? If time permits, I would like to see a simple experiment on thise on any of the datasets (which ever is faster to train/test). Please reach out to me if you need more clarification on this.

7. **Inference Time**: Can the authors provide comparison between all the baselines and RSM with regards to inference time on the different datasets evaluated?


**[Minor comments which I have not considered for rating of the paper]** -- The authors need not reply to these 2 comments below.

8. I think reconstruction results shown using SSIM or MSE on most simplistic datasets such as OBJ3D or CLEVERER where the values are in the 90s aren't really indicative of how useful the representations are for downstream tasks. It appears that VQA task in CLEVERER is also pretty simplistic and has reached mid-90s -- it would be better the community in general moves to addressing complex tasks of planning, RL and hard-VQA (such as Physion VQA task as shown in the paper).

9. **Readability of methodology section**: In section 2.1 (RSM Overview), it would be helpful to re-iterate what the purpose of having CCI is. The only mention of CCI is in the last section of introduction, so explicitly having it as a part of the RSM overview would be helpful.


----
**Rationale for rating**
I find the contributions of this work to be novel and experimentation to be through and hence I lean towards accepting the paper. However, I would like the authors to address my questions before I finalize my decision. In addition to this, if authors have clarifying questions regarding comment (6), I would request them to reach out to me early on during the rebuttal phase.

**Questions:**

See *Weaknesses*

---

> ### Author Rebuttal · Authors · 2023-08-10
>
> Thank you for your valuable reviews. We deeply appreciate the time and effort you invested in evaluating our work.
>
>
> We are happy to see `intuitive and novel`, `significant improvement`, and `impressive`  in your reviews.
>
>
>
> >3.  **CCI**: **Have authors tried treating each slot independently as input to the transformer with positional encoding instead of concatenating past  $T$  time slots?**
>
> We have not tried doing so, and a major reason was the observation in one of our baselines, SlotFormer [1]. There they explicitly pass slots as tokens to a transformer and positional encoding is added temporally across the previous $T$ steps, yet even with positional encoding, they concatenate all slots across the span of $T$ frames as input to their transformer. Given their success with concatenation, we also opted for doing so, and did not use positional embedding alone.
>
> Finally, we prioritized our compute budget during the rebuttal phase to address your concern wrt comment 6. However, we would certainly explore the effect of using positional encoding alone vs. concatenation before the final version.
>
>
> ---
>
> > 4.  **In Fig 1(c), is should the  $t$  in blue circle be  $s_t$.**
>
> Thank you for your question, and sorry for the confusion on $s_t$ notation in Figure 1c.
>
> We had opted for a notation where any colored circle with a solid border represents a particular slot at the time that is given inside the circle (i.e., $t$ or $t+1$), but you are right, and it can be confusing when such slots appear alongside $\Delta s_t$, we should update all colored circles with solid borders to contain symbols such as $s^i_t$, denoting slot number and its time.
>
> We have updated Figure 1 in our latest version. Thank you.
>
> ---
>
> > 5.  **Memory Constraints**: The CCI (Step 1) has a huge memory constraint of  ($\tau$+1)×$N$×$d_s$  which is  7×6×128=5376  for OBJ3D dataset (which has the smallest memory requirements), and  16×6×192=18432  which is huge.
>
> Thank you for your helpful observation.
>
> We would like to mention that this memory constraint could, to some extent, be addressed by choosing $N$ more wisely. One way, when it comes to more realistic scenes, is to segment a scene comprising numerous objects into patches (with possible overlapping) where the objects inside the patch are unlikely to interact with those outside; this will keep the dependence on $N$ to a constant rate.
>
> Lastly, we would appreciate it if the reviewer could help us realize in which settings our method could constitute a potential memory-related problem since even with batches as big as 512, the memory consumption of this block seems to be ~100MB at most, which doesn't cause a critical hurdle in our experiments. One possible future work to try is to build a bank of memory and use the memory retrieval method as in Goyal et al. ICLR 2022, in which, in each step, appropriate information from the large memory pool can be retrieved as a context. We would appreciate the reviewer's insight in case we've missed an angle.
>
> ---
>
> > 6.  **OOD in dynamics**: How does RSM behave when it sees an OOD dynamics for a particular object.
>
> Thank you for your insightful suggestion regarding the  OOD  evaluation strategy. We conducted additional experiments for OOD dynamics and presented our approach and findings in response to your question. **Please refer to  Figure 4 in the supporting document for visualizations and Section "3. Additional evaluation in OOD dynamics" in the Global Rebuttal for more details**.
>
>
> In conclusion, our findings underscore RSM's potential to generalize and reuse objects' motion mechanisms across different colors. However,  there are still limitations in mechanism prediction in OOD scenarios, particularly when confronted with strong OOD cases, as blue balls appear only in the test set.
>
>
>
>
>
> ---
> > 7.  **Inference Time**: Can the authors provide comparison between all the baselines and RSM with regards to inference time on the different datasets evaluated?
>
>
> We conduct the analysis to compare the inference duration (in seconds) among methods on the same 1 Tesla V100-SXM2 GPU. The reported values are the average inference time per batch, where the average is taken over the batches in the test set. The batch size is the same within one dataset but could be different across the different datasets.
>
>
> | Method         | OBJ3D                 | CLEVRER               | Physion               | PHYRE                 |
> |----------------|-----------------------|-----------------------|-----------------------|-----------------------|
> | **SlotFormer** | 0.1655 ± 0.012       | 0.1889 ± 0.009       | 0.0349 ± 0.014       | 0.5322 ± 0.017       |
> | **SwitchFormer** | 0.2636 ± 0.007       | 0.2634 ± 0.011       | 0.1336 ± 0.009       | 0.6334 ± 0.012       |
> | **NPS**        | 0.1639 ± 0.020       | 0.1877 ± 0.014       | 0.0807 ± 0.005       |  **0.4733** ± 0.010    |
> | **RSM**        |  **0.1526** ± 0.014   |  **0.1833** ± 0.016  |  **0.0324** ± 0.012   | 0.4794 ± 0.020       |
>
> According to the table, we can see that RSM takes the shortest duration for inference for most of the datasets or is on par with the rest of the baselines in the rest of the datasets.
> However, it is worth mentioning that the part which accounts for most of the inference time is the sequential prediction of rollout steps (that all models encounter), not the prediction of slots' next step (which is different among models).
>
>
> ---
> [1] SlotFormer: Unsupervised Visual Dynamics Simulation with Object-Centric Models - Ziyi Wu, Nikita Dvornik, Klaus Greff, Thomas Kipf, Animesh Garg

---

> > ### Comment · Reviewer_NsYp · 2023-08-19
> > **Thanks for the rebuttal**
> >
> > Thank you to the authors for their rebuttal and showing some more visualizations on OOD as requested. The Fig 4 in the posted pdf during rebuttal sheds potentially important light on what object-centric dynamics models are (and are not) capable of in their current form. The oddly predicted shape of the blue ball during the OOD rollout is interesting example.
> >
> > Also, based on reviewers DY7c and 8CU7's comments -- I appreciate adding the **MOVi-C**  dataset. As the authors are aware of it -- the model does not seem to be tune correctly yet as the visual results are blurry.
> >
> > > Lastly, we would appreciate it if the reviewer could help us realize in which settings our method could constitute a potential memory-related problem since even with batches as big as 512,...
> >
> > Sure. My concern was if RSM-like model is used in say model-based RL framework such as Dreamer -- because the policy learning happens in imagination (via rollout of the world model) -- it might be additionally expensive to do both `forward` and `backward` passes on the model.
> > But given the authors response on memory and the inference speed (forward pass only) -- I think my concerns have been alleviated.
> >
> > Thanks to the authors once again. I will change my score post-reviewer + AC discussion period.

---

> > > ### Author Response · Authors · 2023-08-20
> > >
> > > Thank you for carefully reviewing our rebuttal and for your consideration of changing the score.
> > >
> > > We're glad that our rebuttal alleviated your concerns and very appreciate your discussion about the scenario when applying an RSM-like model in RL.

---

### Official Review · Reviewer_DY7c · 2023-07-13

**Soundness:** 3 good
**Presentation:** 3 good
**Contribution:** 3 good
**Rating:** 6
**Confidence:** 2

**Summary:**

This work presents a novel framework to model object dynamics by leveraging communication slots in a modular architecture. The method uses Central Contextual Information (CCI) to allow information exchange among existing slots. The authors demonstrate the efficacy of their method's superior empirical performance on VQA, action planning, and future frame prediction.

**Strengths:**

1. The idea of using central contextual information is novel and interesting and seems to provide clear empirical benefits over prior work.
2. The authors have clearly outlined the primary contributions and have carefully designed ablations to support their claim.

**Weaknesses:**

One question I have for the authors is how this method would be extended to a real scene where the entities that are interacting cannot be clearly separated. For instance, in CLEVRER or Obj3D, it's known ahead of time that the 3-5 objects would be moving around and interacting with each other. But in a real scene, finding that might get very challenging. I would be interesting in hearing author's perspective about that.

**Questions:**

See weaknesses.

**Limitations:**

See weaknesses.

---

> ### Author Rebuttal · Authors · 2023-08-10
>
> Thank you for your valuable reviews. We deeply appreciate the time and effort you invested in evaluating our work.
>
> We are glad to see `novel and interesting`, `clear empirical benefits`, `clearly outlined contributions`, and `carefully designed ablations` in your reviews.
>
> ---
> > **One question I have for the authors is how this method would be extended to a real scene where the entities that are interacting cannot be clearly separated.** For instance, in CLEVRER or Obj3D, it's known ahead of time that the 3-5 objects would be moving around and interacting with each other. But in a real scene, finding that might get very challenging. I would be interesting in hearing author's perspective about that.
>
> We appreciate the insightful comment and acknowledge it is a valid concern. However, we would like to emphasize that RSM is particularly suited to be extended to such scenarios, the reason being that RSM works directly with slots and does not care about their number. And the problem of not knowing the number of objects beforehand has been studied in the literature. Most notably,  going towards more realistic environments, we would use superior object-centric architectures such as Slot Attention [1], SAVi, SAVi++ [2,3], where they have successfully scaled to real-world self-driving datasets [3]. So the task of object discovery in real scenes has been extensively studied, and with enough computing, we could scale to real-world scenes as well. Regarding the variable number of objects, please refer to [4] where a Dirichlet Process Mixture Model is employed for adaptively changing the number of mixtures required to represent the observed data in a continual learning setting. Since the methods in [1,2,3] basically constitute a learnable clustering scheme, we believe [4] could be a reasonable candidate for adaptively choosing the number of such clusters/mixtures (in our case, objects) in real scenes.
>
> Turning our attention to an extension of our work on a more complex dataset, we present additional experimental results from the MOVi-C dataset—a rich repository for video prediction featuring realistic scenes and intricate interactions.
> **Please refer to   Figure 1 in the supporting document for visualizations and the "1. Additional experiment on the realistic dataset - MOVi-C" section in Global Rebuttal**.
>
>
>
> We hope our explanations and the initial experimental results in the more realistic datasets will help address your concerns and help you with your assessment.
>
> ---
> [1] Object-Centric Learning with Slot Attention - Francesco Locatello, Dirk Weissenborn, Thomas Unterthiner, Aravindh Mahendran, Georg Heigold, Jakob Uszkoreit, Alexey Dosovitskiy, Thomas Kipf
>
> [2] SAVI Conditional Object-Centric Learning from Video - Thomas Kipf, Gamaleldin F. Elsayed, Aravindh Mahendran, Austin Stone, Sara Sabour, Georg Heigold, Rico Jonschkowski, Alexey Dosovitskiy, Klaus Greff
>
> [3] SAVi++: Towards End-to-End Object-Centric Learning from Real-World Videos - Gamaleldin F. Elsayed, Aravindh Mahendran, Sjoerd van Steenkiste, Klaus Greff, Michael C. Mozer, Thomas Kipf
>
> [4] A Neural Dirichlet Process Mixture Model for Task-Free Continual Learning - Soochan Lee, Junsoo Ha, Dongsu Zhang, Gunhee Kim

---

### Author Rebuttal · Authors · 2023-08-10

We would like to say thank all reviewers for their time and efforts to review our work.

The following sections are explanations related to the one-page PDF visualization.

The outline is as follows:
1. **Additional experiment on the realistic dataset - MOVi-C** as suggested by reviewers DY7c and 8CU7.
2. **Mechanisms assignment in OBJ3D and CLEVRER** as suggested reviewer 8CU7.
3. **Additional evaluation in OOD dynamics** as suggested by reviewer TEgG.



---
### 1. Additional experiment on the realistic dataset - MOVi-C

**Please refer to   Figure 1 in the PDF for visualizations**

The following is a quick summary of our configurations on MOVi-C:
- Dataset: Training was conducted on approximately 9000 videos, with testing performed on 1000 videos.
- Object-centric Training: We employed a StoSAVi (a stochastic version of SAVi [1], introduced in SlotFormer [2]) model equipped with 7 slots, trained over 50 epochs.
- RSM Training: Our RSM model underwent 20 epochs of training, utilizing 10 burn-in frames as input and predicting 14 rollout frames.




According to the initial experiment result in **Figure 1**,  RSM exhibited the ability to manage complex scenes and interactions, as showcased in the MOVi-C dataset. An example is in RSM's accurate prediction of the dynamics, such as the motion of the white can to the right and the downward trajectory of the lion toy.

Due to the time limitations for the rebuttal period, we cannot work on a larger dataset or fine-tune both object-centric and RSM models carefully, thus causing low-quality visuals such as losing small objects and object details. Nonetheless, it is worth noticing the potential of RSM in modeling dynamics in complex scenes from these initial results. These initial results also motivate us to further work on this family of realistic datasets.

[1] SAVI Conditional Object-Centric Learning from Video - Thomas Kipf, Gamaleldin F. Elsayed, Aravindh Mahendran, Austin Stone, Sara Sabour, Georg Heigold, Rico Jonschkowski, Alexey Dosovitskiy, Klaus Greff
[2] SlotFormer: Unsupervised Visual Dynamics Simulation with Object-Centric Models - Ziyi Wu, Nikita Dvornik, Klaus Greff, Thomas Kipf, Animesh Garg


---
### 2. Mechanisms assignment in OBJ3D and CLEVRER
**Please refer to Figures 2 and 3 in the PDF for visualizations**

According to the mechanism assignments in the analysis, we could see:
- It is interesting to see that the environments share similar motions and interactions as OBJ3D and CLEVRER share a similar set of mechanisms, such as `Moving and spinning`, `Moving fast`, and `Moving slow`.
- With some complex motions like `Moving and spinning`, a separated mechanism is appointed (by the model during training).

Below, we provide the ratio of selected mechanisms over the test set for reference:
| Inferred Mechanism | OBJ3D | CLEVRER|
|--|--|--|
|**Idle** | 0.435 | 0.370
|**Moving Fast** | 0.107 | 0.212
|**Moving Slow** | 0.064 |  0.129
|**Collide** |0.096 | 0.076
|**Moving and Spinning** | 0.045|  0.047
|**Redundant Mechanisms** |0.249 | 0.163

The Redundant Mechanisms (2 mechanisms in particular) we believe are artifacts of having more mechanisms and slots employed than there are needed for the environment. We observe that these mechanisms usually attend to slots that contain no object, and in fact, removing such mechanisms only affects the performance very slightly, further confirming their redundancy, and this artifact could be expected because we do not tune the number of slots and mechanisms per each environment. The rest of the mechanisms demonstrate meaningful transitions, as can be seen in the new visualizations.


---
### 3. Additional evaluation in OOD dynamics
**Please refer to   Figure 4 in the PDF for visualizations**


We introduce modifications to the PHYRE dataset, specifically altering the characteristics of blue objects between the train and test sets as follows:
- **Modified train set**: Blue objects in the train set are **static** since they represent blue floors or walls.
- **Modified test set**: Blue objects in the test set are allowed to **move**, simulating scenarios involving blue balls or glasses.

Particularly,  our approach involves extracting templates containing blue balls and glasses from the training set and transferring them to the modified test set. Notice that the modified train set includes the ball objects in red, green, white, and gray colors, as well as the glass objects in gray color. In PHYRE, it's impossible to have, for instance, a red object that is NOT a ball to be static in the dataset.



In response to the question:" `Would RSM reuse the mechanism corresponding to the motion of the object?`", we offer the following insights:

- **Yes**. RSM exhibits the capacity to reuse mechanisms based on object motion appropriately. For instance, the blue ball is assigned the "**Fall**" mechanism from steps 0 to 2, the "**Collide**" mechanism from steps 2 to 3, and the "**horizontal moving**" mechanism (leftward) in the subsequent steps. These mechanism assignments effectively capture the object's dynamics compared to the target frames.
- RSM demonstrates the ability to generalize object motion across different colors and the ability to reuse mechanisms in OOD cases by assigning appropriate mechanisms to the blue ball, corresponding to its motions over steps (falling, colliding, and moving left).
- However, since the blue balls appear only in the test set could be considered a strong OOD case (the model has not had a chance even to reconstruct a blue ball-shaped object); the predicted next frames exhibit some limitations in comparison to predictions trained on the standard training set, such as distortions in the shape of the blue ball and incorrect position of the red ball from step 4 after the collision of the blue and red balls.

---

### Decision · Program_Chairs · 2023-09-21

**Decision:**

Accept (poster)

**Comment:**

This paper received mixed reviews in the initial round of review. However, the authors have done a very good job addressing most of the comments and points of feedback. Given this effective rebuttal, there was a consensus by all reviewers to accept this paper. I have read the reviews and the rebuttal, and given these I recommend accepting the paper.